# Formation of large low shear velocity provinces through the decomposition of oxidized mantle

Wenzhong Wang [1,2✉], Jiachao Liu[3✉], Feng Zhu [4], Mingming Li [5], Susannah M. Dorfman [3], Jie Li [4] & Zhongqing Wu[1,6,7✉]

Large Low Shear Velocity Provinces (LLSVPs) in the lowermost mantle are key to understanding the chemical composition and thermal structure of the deep Earth, but their origins have long been debated. Bridgmanite, the most abundant lower-mantle mineral, can incorporate extensive amounts of iron (Fe) with effects on various geophysical properties. Here our high-pressure experiments and ab initio calculations reveal that a ferric-iron-rich bridgmanite coexists with an Fe-poor bridgmanite in the 90 mol% $MgSiO_3$–10 mol% $Fe_2O_3$ system, rather than forming a homogeneous single phase. The $Fe^{3+}$-rich bridgmanite has substantially lower velocities and a higher $V_P/V_S$ ratio than $MgSiO_3$ bridgmanite under lowermost-mantle conditions. Our modeling shows that the enrichment of $Fe^{3+}$-rich bridgmanite in a pyrolitic composition can explain the observed features of the LLSVPs. The presence of $Fe^{3+}$-rich materials within LLSVPs may have profound effects on the deep reservoirs of redox-sensitive elements and their isotopes.

[1] Laboratory of Seismology and Physics of Earth's Interior, School of Earth and Space Sciences, University of Science and Technology of China, Hefei, China. [2] Department of Earth Sciences, University College London, London, UK. [3] Department of Earth and Environmental Sciences, Michigan State University, East Lansing, MI, USA. [4] Department of Earth and Environmental Sciences, University of Michigan, Ann Arbor, MI, USA. [5] School of Earth and Space Exploration, Arizona State University, Tempe, AZ, USA. [6] National Geophysical Observatory at Mengcheng, University of Science and Technology of China, Hefei, China. [7] CAS Center for Excellence in Comparative Planetology, USTC, Hefei, Anhui, China. ✉email: wz30304@mail.ustc.edu.cn; jiacliu09@gmail.com; wuzq10@ustc.edu.cn

The large low shear velocity provinces (LLSVPs) are two massive and mysterious regions sitting beneath Africa and the Pacific[1–3] and occupy ~3–9% of the volume of the Earth[3,4]. They are characterized by their lower-than-average seismic wave velocities[4] and extend by thousands of kilometers laterally and up to >1000 km vertically above the core–mantle boundary (CMB)[3,4]. As the largest seismic heterogeneities in the lower mantle, they may hold the key to understanding the thermal, chemical, and dynamical evolution of the Earth[5,6,7]. Different shear-wave tomography models[4] have reached agreement that the shear-wave velocity anomaly ($dln V_S$) ranges from −0.5 to −1.0% in the shallow part of LLSVPs, while it could be up to −3.0% in the bottom part[6]. The compressional-wave tomography models also reveal negative anomalies of compressional-wave velocities ($V_P$) (refs. [8,9]), although the amplitude, shape, and geographical location of $V_P$ anomaly vary widely among different models[5]. The $V_P$ anomaly generally has a smaller amplitude than the $V_S$ anomaly, causing a high $dln V_S/dln V_P$ ratio[9]. In particular, waveform and travel-time seismic studies[1,3,10–12] reveal that the LLSVPs have sharp edges along their margins, which is also supported by the large lateral $dV_S$ gradients at the boundary of LLSVPs[6].

The systematic and discontinuous contrasts in seismic properties indicate that the LLVSPs are likely composed of distinct chemical materials from the surrounding mantle[11,13]. A chemically distinct origin of the LLSVPs may be implicated by their density anomalies as well; however, large discrepancies for the density anomaly associated with the LLSVPs exist in the literature[5]. Recent tidal tomography based on body tide displacements[14] found that the mean density of the lower two-thirds of the two LLSVPs is ~0.5% higher than that of the surrounding mantle. On the contrary, a study using Stoneley modes suggested an overall negative density anomaly within LLSVPs, without excluding the possibility of a high-density anomaly within the lowermost LLSVPs[15]. It is still unknown whether the regional differences in density anomaly are caused by the choice of observations used to constrain density models or reflect the nature of LLSVPs associated with their origins.

Hypotheses for the origin of chemically distinct LLSVPs include processes associated with the accumulation of subducted oceanic crust over Earth history[16] and the differentiation and solidification of an ancient basal magma ocean[6]. Sunken piles of subducted oceanic crust, which is compositionally different from and significantly denser than the pyrolitic lower mantle[17], was proposed to explain LLSVPs because of the low velocity of calcium silicate perovskite (CaSiO₃, CaPv) (ref. [18]). However, there are significant discrepancies in the velocity of CaPv between two experimental studies[18,19] and between experiments and theoretical results[20]. Elastic properties from ab initio simulations for the entire MORB assemblage indicate that subducted oceanic crust has relatively higher velocities than the ambient mantle[21]. Moreover, geodynamic simulations[22] suggested that the present-day subducted oceanic crust is too thin to provide enough negative buoyancy to survive viscous stirring and it hence is difficult to amass coherent thermochemical structures and shapes at the CMB similar to LLSVPs[23]. Alternatively, LLSVPs may be composed of primordial residues from basal magma ocean crystallization or core–mantle differentiation that have not yet been fully homogenized by the mantle convection[24–26]. These primordial materials would need to be intrinsically more dense than the surrounding mantle to overcome mantle stirring[27]. Dense Fe–Ni–S liquid, for instance, was proposed to explain the LLSVPs[28], but the amount of this liquid remaining in the deep mantle, which depends on the drainage of melt to the core[29], is under debate.

Consistent with both efficient drainage of metallic melt and a primordial origin of the LLSVPs is chemical heterogeneity produced by redox reactions in the magma ocean. Ferrous iron ($Fe^{2+}$) in silicate melts has been observed to disproportionate to ferric iron ($Fe^{3+}$) plus metallic iron ($Fe^{0}$) at high pressures[30]. Segregation of precipitated $Fe^{0}$ from the magma ocean into the core would enrich $Fe^{3+}$ in the mantle. Bridgmanite (Bdg), the dominant $Fe^{3+}$-bearing mantle mineral, hosts $Fe^{3+}$ through the $Fe^{3+}$–$Fe^{3+}$ or $Fe^{3+}$–$Al^{3+}$ charge-coupled substitution in the deep mantle[31–35]. In particular, a $Fe^{3+}$-rich Bdg with the chemical composition of $(Mg_{0.5}Fe_{0.5})(Si_{0.5}Fe_{0.5})O_3$ was recently synthesized by Liu et al.[36]. Oxidized domains enriched with such $Fe^{3+}$-rich Bdg would be distinct from a pyrolitic lower mantle in sound velocity and density[37,38] and may be responsible for the origins of lower-mantle seismic and geochemical heterogeneities, such as the LLSVPs[1–3,6]. The conditions of formation of such $Fe^{3+}$-rich Bdg in a mantle phase assemblage and its elastic properties at lower-mantle-relevant temperatures are vital to test this hypothesis, but these questions remain unclear.

In this work, we combine high pressure-temperature ($P$-$T$) experiments, ab initio calculations, and geodynamic simulations to study the formation of $Fe^{3+}$-rich Bdg, its thermoelastic properties, and its dynamics in the mantle, to evaluate whether enrichment in $Fe^{3+}$-rich Bdg can explain the seismic signatures of LLSVPs.

## Results

### The coexistence of $Fe^{3+}$-rich and Fe-poor bridgmanite phases.
We conducted a series of multi-anvil experiments with the bulk composition of 90 mol% MgSiO₃–10 mol% Fe₂O₃ from 10-24 GPa along a mantle geotherm (Supplementary Table 1). At 10 GPa and 1573 K (Fig. 1d), the run products consist of separate MgSiO₃ and Fe₂O₃ phases, demonstrating low solubility of Fe₂O₃ in clinopyroxenes at upper-mantle conditions due to the incompatibility between large $Fe^{3+}$ and small tetrahedral site. At 15 GPa, an iron-rich akimotoite (Aki) forms with 33 mol% of MgSiO₃ and 67 mol% of Fe₂O₃, coexisting with $(Mg_{1.79}Fe_{0.18})SiO_4$ wadsleyite and SiO₂ stishovite (Fig. 1c). This indicates that the iron-rich Aki coexisting with iron-depleted oxides/silicates are energetically more stable than a single-phase MgSiO₃–Fe₂O₃ solid solution with intermediate iron content. At 24 GPa and 1873 K, which corresponds to the $P$–$T$ conditions around the uppermost lower mantle, the products consist of Fe-poor Bdg and Fe-rich Aki with 44–49 mol% Fe in both Mg and Si sites (Fig. 1a and Supplementary Table 1), instead of forming a single phase of $(Mg_{0.9}Fe_{0.1})(Si_{0.9}Fe_{0.1})O_3$ Bdg. The Fe-rich Aki is evenly distributed in the matrix of the Fe-poor Bdg (Fig. 1a). The run products of experiments running at the same $P$–$T$ conditions for 8 and 24 h have the same compositions within analytical uncertainty (Supplementary Table 1), confirming that the experiments reached equilibrium. In situ XRD measurements coupled with a diamond anvil cell (DAC) show that this Fe-rich Aki phase transforms to Bdg phase at 23.5 ± 1.0 GPa and 300 K (Supplementary Fig. 1). Moreover, this $Fe^{3+}$-rich Bdg phase completely transforms back to Aki with the same lattice parameters as the starting Aki phase after the decompression of the DAC (Supplementary Fig. 1). The reversible phase transition of this Fe-rich phase means that for our multi-anvil experiments at the $P$–$T$ conditions of the uppermost lower mantle, $Fe^{3+}$-rich Bdg coexists with Fe-pool Bdg.

A previous multi-anvil study[39] synthesized $Fe^{3+}$-only bridgmanite with 2–4 mol% $Fe^{3+}$ in the cation sites but did not observe the $Fe^{3+}$-rich Bdg phase, possibly because the bulk Fe content of their experiments is not high enough to enable the formation of such $Fe^{3+}$-rich Bdg. Moreover, the presence of unreacted MgO and SiO₂ in their run products (Fig. 1 in ref. [39]) suggests that their starting materials may not be homogenous or

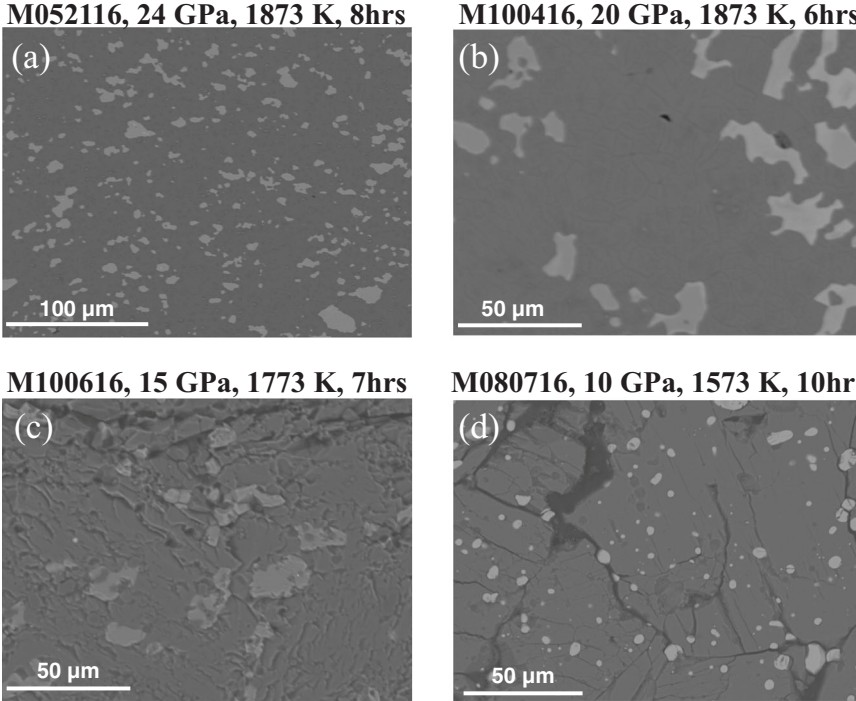

**M052116, 24 GPa, 1873 K, 8hrs**

(a)

100 μm

**M100416, 20 GPa, 1873 K, 6hrs**

(b)

50 μm

**M100616, 15 GPa, 1773 K, 7hrs**

(c)

50 μm

**M080716, 10 GPa, 1573 K, 10hrs**

(d)

50 μm

**Fig. 1 Back-scattered scanning electron microscope images of recovered experimental phase assemblages forming from 90% MgSiO₃ + 10% Fe₂O₃ starting mixture after equilibrium at mantle pressure–temperature conditions for 6–10 h.** Bridgmanite/akimotoite with ~50 mol% $Fe_2O_3$ substitution in both Mg and Si sites (bright phases in (**a–c**)) coexists with normal mantle minerals such as Fe-poor bridgmanite (dark phases in (**a**) and (**b**)) at >20 GPa, or with wadsleyite (gray phase in (**c**)) + stishovite (dark phase in (**c**)) at 15 GPa, while $Fe_2O_3$ (bright phase in (**d**)) was stabilized as a separate phase with Fe-depleted clinopyroxene (dark phase in (**d**)) at 10 GPa.

their experiments did not reach chemical equilibrium. Another study[40] synthesized $Fe^{3+}$-only Bdg with the starting material of 90 mol% $MgSiO_3$–10 mol% $Fe_2O_3$ using laser-heated diamond anvil cell (LH-DAC)[40]. However, the chemical composition of $Fe^{3+}$-only Bdg was not reported, possibly because there was a significant loss of Fe and Mg during melting[40], and some $Fe^{3+}$ was reduced through reaction with diamond during laser heating[41]. In addition, the proportion of the Fe-rich Bdg phase is much smaller than the Fe-poor Bdg (Fig. 1), and therefore it is difficult to detect without a detailed analysis of the run products in ref. [40].

We also performed ab initio calculations (see methods and supplementary materials) to investigate the stability of $(Mg_{0.5}Fe_{0.5})(Si_{0.5}Fe_{0.5})O_3$ Bdg under lower-mantle conditions. Our results show that the assemblage of $(Mg_{0.5}Fe_{0.5})(Si_{0.5}Fe_{0.5})O_3$ and $MgSiO_3$ Bdg has a lower Gibbs free energy than a single-phase $(Mg_{0.875}Fe_{0.125})(Si_{0.875}Fe_{0.125})O_3$ Bdg under the P–T of the whole lower mantle regardless of the spin state (Supplementary Fig. 2), indicating that the mixed two phases are more stable than the single-phase Bdg with a homogeneous composition. The theoretical results support our experimental observations and reveal that this $Fe^{3+}$-rich Bdg with the chemical composition of approximately $(Mg_{0.5}Fe_{0.5})(Si_{0.5}Fe_{0.5})O_3$ should form as a separate phase coexisting with Fe-poor Bdg in the bulk composition of 90 mol% $MgSiO_3$–10 mol% $Fe_2O_3$ due to the miscibility gap.

**Elastic properties and sound velocities of $(Mg_{0.5}Fe_{0.5})(Si_{0.5}Fe_{0.5})O_3$ bridgmanite.** Determining the seismic signature of a separate $Fe^{3+}$-rich phase in equilibrium with the mantle phase assemblage requires elastic properties of this phase as a function of pressure and temperature conditions in the mantle. The elastic properties of $(Mg_{0.5}Fe_{0.5})(Si_{0.5}Fe_{0.5})O_3$ Bdg up to 130 GPa and 3000 K were theoretically obtained from ab initio calculations.

Because Bdg may also accommodate Al in the octahedral site[31,38,42], we also conducted calculations on an end-member composition $(Mg_{0.5}Fe_{0.5})(Si_{0.5}Al_{0.5})O_3$, to quantify the effect of Al on the elasticity of Bdg. In these calculations, the octahedral site (B-site) $Fe^{3+}$ in $(Mg_{0.5}Fe_{0.5})(Si_{0.5}Fe_{0.5})O_3$ Bdg undergoes a high-spin (HS) state to a low-spin (LS) state transition with increasing pressure (Fig. 2a), while the dodecahedral-site (A-site) $Fe^{3+}$ in both compositions remain in the HS state throughout the lower-mantle conditions[43].

Our calculated volumes of HS- and LS-$(Mg_{0.5}Fe_{0.5})(Si_{0.5}Fe_{0.5})O_3$ Bdg agree well with experimental measurements at 300 K (ref. [36]) (Fig. 2a). The calculated spin transition of the B-site $Fe^{3+}$ occurs between 49 and 55 GPa at 300 K[36], which is slightly higher and narrower than experimental results[36]. The predicted volume collapse ($\Delta V^{HS-LS}$) caused by the spin transition of B-site $Fe^{3+}$ is ~4.3% at 300 K, higher than experimental measurements (2.7%)[36] but consistent with previous theoretical calculations[44] on $(Mg_{0.5}Fe_{0.125})(Si_{0.5}Fe_{0.125})O_3$ Bdg assuming that $\Delta V^{HS-LS}$ is linearly dependent on $Fe^{3+}$ content. The $\Delta V^{HS-LS}$ discrepancy between experimental and theoretical studies is probably caused by the difference in the pressure range for the mixed-spin (MS) state. The predicted volumes of $(Mg_{0.5}Fe_{0.5})(Si_{0.5}Al_{0.5})O_3$ Bdg also show excellent agreement with experimental results at 300 K[42]. These comparisons demonstrate the high reliability of our DFT + U calculations in predicting elastic properties, as suggested by previous studies[43–45].

The spin transition of B-site $Fe^{3+}$ in $(Mg_{0.5}Fe_{0.5})(Si_{0.5}Fe_{0.5})O_3$ Bdg generates a strong effect on bulk modulus ($K_S$) and $V_P$, which both show deep valleys that broaden and decrease in magnitude with increasing temperature (Supplementary Fig. 3). The magnitude and width of the $K_S$ and $V_P$ anomalies are controlled by the fraction of LS B-site $Fe^{3+}$ ($n_{LS}$) and the pressure and temperature dependences of $n_{LS}$ (see Supplementary Materials). Compared to $MgSiO_3$ Bdg[46], both $(Mg_{0.5}Fe_{0.5})(Si_{0.5}Fe_{0.5})O_3$ and

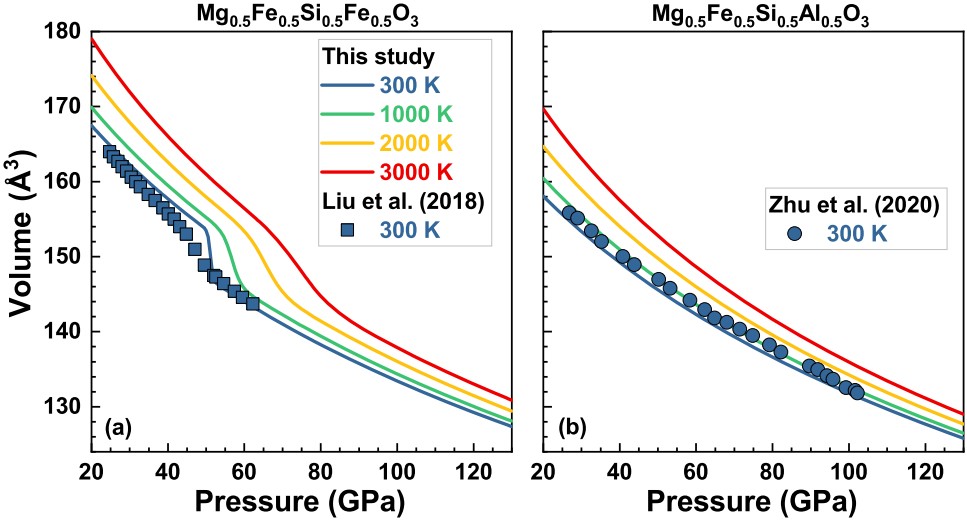

**Fig. 2 Isothermal compression curves predicted by ab initio calculations for ($Mg_{0.5}Fe_{0.5}$)($Si_{0.5}Fe_{0.5}$)$O_3$ and ($Mg_{0.5}Fe_{0.5}$)($Si_{0.5}Al_{0.5}$)$O_3$ bridgmanite.**
**a** ($Mg_{0.5}Fe_{0.5}$)($Si_{0.5}Fe_{0.5}$)$O_3$ bridgmanite, **b** ($Mg_{0.5}Fe_{0.5}$)($Si_{0.5}Al_{0.5}$)$O_3$ bridgmanite. The blue, green, orange, and red curves are calculated compression curves of ($Mg_{0.5}Fe_{0.5}$)($Si_{0.5}Fe_{0.5}$)$O_3$ and ($Mg_{0.5}Fe_{0.5}$)($Si_{0.5}Al_{0.5}$)$O_3$ bridgmanite at 300, 1000, 2000, and 3000 K, respectively. The blue squares are experimental measurements from Liu et al.[36], which shows that the spin transition of $Fe^{3+}$ in the B-site of ($Mg_{0.5}Fe_{0.5}$)($Si_{0.5}Fe_{0.5}$)$O_3$ bridgmanite occurs between 43 and 53 GPa at 300 K. The blue circles are experimental results of Zhu et al.[42].

($Mg_{0.5}Fe_{0.5}$)($Si_{0.5}Al_{0.5}$)$O_3$ Bdg have much lower elastic moduli ($K_S$ and $G$) and velocities ($V_P$ and $V_S$) (Supplementary Fig. 4). At lowermost-mantle conditions, the differences in $K_S$, $G$, $V_P$, and $V_S$ between ($Mg_{0.5}Fe_{0.5}$)($Si_{0.5}Fe_{0.5}$)$O_3$ and $MgSiO_3$ Bdg[46] are about −5%, −37%, −17%, and −28%, respectively, which in turn causes a higher $V_P/V_S$ ratio of 2.1 in ($Mg_{0.5}Fe_{0.5}$)($Si_{0.5}Fe_{0.5}$)$O_3$ Bdg (Supplementary Fig. 4). By comparison, the differences in $K_S$, $G$, $V_P$, and $V_S$ between ($Mg_{0.5}Fe_{0.5}$)($Si_{0.5}Al_{0.5}$)$O_3$ and $MgSiO_3$ Bdg are about −4%, −18%, −9%, and −14%, respectively.

## Discussion

Combining elastic data from previous studies[20,45–47] with our results, we modeled the density and velocity anomalies caused by the presence of $Fe^{3+}$-rich Bdg relative to the pyrolitic composition, which can effectively reproduce the reference seismic velocities and density of PREM[37,38]. The modeled chemical assemblage has pyrolitic mineral fractions (15% ferropericlase (Fp) + 78% Bdg + 7% CaPv) in which a portion of $Fe^{2+}$-bearing Bdg was substituted by ($Mg_{0.5}Fe_{0.5}$)($Si_{0.5}Fe_{0.5}$)$O_3$ Bdg. We find that the enrichment of ($Mg_{0.5}Fe_{0.5}$)($Si_{0.5}Fe_{0.5}$)$O_3$ Bdg in the assemblage can explain the seismic features of the LLSVPs[4,6,8,9]. The $V_S$ anomalies of −1.5% to −3.0% and the large $dlnV_S/dlnV_P$ ratio >2.0 observed in LLSVPs[4,6,8,9] can be reproduced by the enrichment of 10–15% ($Mg_{0.5}Fe_{0.5}$)($Si_{0.5}Fe_{0.5}$)$O_3$ Bdg in pyrolite at 110 GPa (Fig. 3). If LLSVPs are hotter ($\Delta T_{LLSVPS} > 0$), the required proportion of ($Mg_{0.5}Fe_{0.5}$)($Si_{0.5}Fe_{0.5}$)$O_3$ Bdg would accordingly decrease; for example, it will decrease by ~2% if $\Delta T_{LLSVPS}$ is +400 K. For a pyrolitic composition, Bdg also contains ~5 mol% $Al_2O_3$, which does not significantly change its velocities and density[48]. When 5 mol% $Al_2O_3$ is incorporated into Bdg, the $V_P$ and $V_S$ anomalies caused by the presence of 15% ($Mg_{0.5}Fe_{0.5}$)($Si_{0.5}Fe_{0.4}Al_{0.1}$)$O_3$ Bdg at $\Delta T_{LLSVPS}$ equal to +400 K will be −1.5% and −3.1% (Fig. 3), respectively, which can also reproduce the $dlnV_S/dlnV_P$ ratio >2.0.

In addition, we find that the required proportion of ($Mg_{0.5}Fe_{0.5}$)($Si_{0.5}Fe_{0.5}$)$O_3$ Bdg increases with the decreasing of $Fe^{2+}$ content in the modeled assemblage (noted by the FeO content in Fp, $Fe^{2+}_{Fp}$) to explain the same $V_P$ and $V_S$ anomalies (Fig. 4). If there is no $Fe^{2+}$, the ($Mg_{0.5}Fe_{0.5}$)($Si_{0.5}Fe_{0.5}$)$O_3$ fraction

that can reproduce the $V_S$ anomaly of −3.0% would be raised to ~17% at $\Delta T_{LLSVPS}$ equal to +400 K (Fig. 4 and Supplementary Fig. 5), which corresponds to a $V_P$ anomaly of ~−1.0% and a $dlnV_S/dlnV_P$ ratio of ~3.0. The $dlnV_S/dlnV_P$ ratio >2.0 tends to be reproduced at relatively low $Fe^{2+}$ contents (Fig. 4). If $Fe^{2+}_{Fp}/Fe^{2+}_{Fp, NM} > 0.9$ ($Fe^{2+}_{Fp, NM}$ is the FeO content of Fp in a normal pyrolitic lower mantle, 18 mol%), the $dlnV_S/dlnV_P$ ratio is less than 2.0. In contrast, when $Fe^{2+}_{Fp}/Fe^{2+}_{Fp, NM}$ is <0.25, the $dlnV_S/dlnV_P$ ratio >2.0 can be reproduced for different $V_S$ anomalies (Fig. 4). Our modeling implies that the enrichment of ($Mg_{0.5}Fe_{0.5}$)($Si_{0.5}Fe_{0.5}$)$O_3$ Bdg in the pyrolite assemblage with relatively lower $Fe^{2+}$ content can explain the velocity anomalies and the $dlnV_S/dlnV_P$ ratio observed in LLSVPs[4,6,8,9].

In contrast to velocity anomalies, the modeled density anomaly ($dln\rho$) could be positive, zero, or negative, depending on the ($Mg_{0.5}Fe_{0.5}$)($Si_{0.5}Fe_{0.5}$)$O_3$ abundance, temperature anomaly, and the fraction of $Fe^{2+}$ in Fp. To produce a $V_S$ anomaly of −3.0% for the assemblage enriched in ($Mg_{0.5}Fe_{0.5}$)($Si_{0.5}Fe_{0.5}$)$O_3$ Bdg, the $dln\rho$ decreases from +1.2% at $\Delta T_{LLSVPS}$ of 0 K and $Fe^{2+}_{Fp}/Fe^{2+}_{Fp, NM}$ of 0.5 to −0.5% at $\Delta T_{LLSVPS}$ of +400 K and $Fe^{2+}_{Fp}/Fe^{2+}_{Fp, NM}$ of 0.0 (Figs. 3, 4 and Supplementary Fig. 5). In general, the density anomaly is correlated with the magnitude of $V_S$ anomalies. For $dlnV_S < −1.0\%$, the $dln\rho$ could be positive if $\Delta T_{LLSVPS}$ is 0 K; however, if $\Delta T_{LLSVPS}$ equals to +400 K, the $dln\rho$ could be positive only when $dlnV_S$ is <−2.5% (Fig. 4). Our modeling suggests that the lowermost parts of LLSVPs with large negative velocity anomalies (<−3.0%)[4,6,8] could be denser than the ambient mantle, while the relatively shallow part of LLSVPs with −0.5 to −1.0% $V_S$ anomalies on average likely have slightly lighter density than the ambient mantle. Recent work proposed that the bottom two-thirds of the two LLSVPs are ~0.5% denser than the surrounding mantle[14], while Koelemeijer et al.[15] argued that the overall density of the LLSVPs is lower than the surrounding mantle. Such different conclusions may be related to different depth sensitivities of the datasets considered[49], regardless of the input observations. Also, geodynamic modeling studies[50,51] suggested that the density anomalies of the LLSVPs relative to the surrounding mantle could be positive near the top and bottom of the LLSVPs but neutral or slightly negative in the middle of the LLSVPs. The density of LLSVPs could also be laterally inhomogeneous due to their

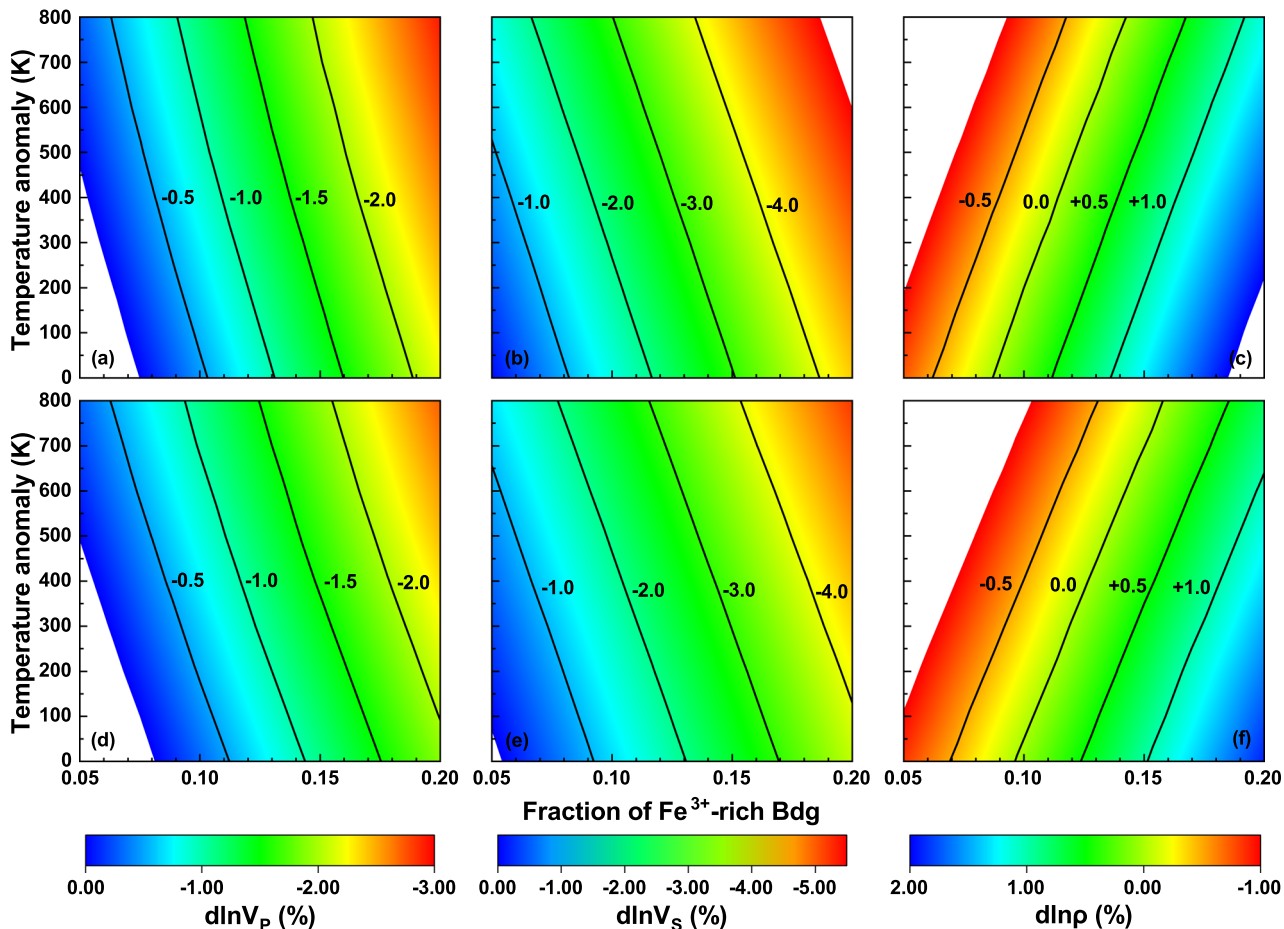

**Fig. 3 Velocity and density anomalies caused by the enrichment of $Fe^{3+}$-rich bridgmanite relative to pyrolitic composition. a**, **d** $V_P$, **b**, **e** $V_S$, and **c**, **f** density anomalies at 110 GPa. **a–c** Anomalies due to enrichment of $(Mg_{0.5}Fe_{0.5})(Si_{0.5}Fe_{0.5})O_3$ bridgmanite (Bdg); **d–f** anomalies due to enrichment of $(Mg_{0.5}Fe_{0.5})(Si_{0.5}Fe_{0.4}Al_{0.1})O_3$ Bdg, where 5% $Al_2O_3$ is incorporated into $(Mg_{0.5}Fe_{0.5})(Si_{0.5}Fe_{0.5})O_3$. Velocities and density of the pyrolitic lower mantle are calculated using the best-fit composition of the lower mantle (15% $Mg_{0.82}Fe_{0.18}O$ ferropericlase (Fp), 78% $Mg_{0.92}Fe_{0.08}SiO_3$ bridgmanite ($Fe^{2+}$-Bdg), and 7% $CaSiO_3$ Ca-perovskite (CaPv))[37]. The modeling chemical assemblage has pyrolitic mineral fractions in which a portion of $Fe^{2+}$-bearing Bdg was substituted by $(Mg_{0.5}Fe_{0.5})(Si_{0.5}Fe_{0.5})O_3$ or $(Mg_{0.5}Fe_{0.5})(Si_{0.5}Fe_{0.4}Al_{0.1})O_3$ Bdg. The initial $Fe^{2+}O$ contents of Bdg and Fp in the modeling assemblage are 4 and 9 mol%, respectively, which are half of those in the reference pyrolite[37]. The temperature anomaly is with respect to the normal mantle temperature from Brown and Shankland[74]. Data for elasticity at high pressure and temperature are derived from previous theoretical studies: Fp, ref. [47]; $Fe^{2+}$-Bdg, ref. [46]; CaPv, ref. [20].

internal convection and the entrainment of multiple compositional components into the LLSVPs[52].

The present model for chemical heterogeneities within the LLSVPs is consistent with Fe-rich remnants of a basal magma ocean created early in Earth's history[6,24–26]. Ferrous Fe in dense silicate melts associated with the basal magma ocean would partially disproportionate to $Fe^{3+}$ plus $Fe^0$ at high pressures[30] and segregation of precipitated $Fe^0$ from the magma ocean into core would enrich silicate melt in $Fe_2O_3$ component. A thermodynamic model of magma ocean crystallization[53] suggests that the silicate melt fraction would be gradually enriched in iron with Fe/(Fe+Mg) >0.3 in the lower mantle after 60 wt% of the melt has solidified. The Fe/(Fe+Mg) ratio in the residual melt remaining in the lowermost mantle could be up to 0.5 near the end of the crystallization. The amount of $Fe^{3+}$ in this melt depends on the amount of $Fe^{2+}$ that would disproportionate into $Fe^{3+}$ plus $Fe^0$ and the efficiency of $Fe^0$ droplet segregation. The required bulk composition with $MgSiO_3$:$Fe_2O_3$ equal to 9:1 could be produced when 40–80% $Fe^{2+}$ undergoes the disproportionation reaction and all $Fe^0$ migrates into the core. $Fe^{3+}$ would be incorporated into bridgmanite with further crystallization, and our experiments and ab initio calculations indicate that in these $Fe^{3+}$-rich regions,

a portion of $(Mg_{0.5}Fe_{0.5})(Si_{0.5}Fe_{0.5})O_3$ silicate would form as a separate phase, coexisting with $Fe^{3+}$-poor silicate. Due to the large excess density, $(Mg_{0.5}Fe_{0.5})(Si_{0.5}Fe_{0.5})O_3$ silicate could descend to the base of the lower mantle through mantle convection and result in $Fe^{3+}$-rich bridgmanite piles. Our geodynamic modeling demonstrates that such $Fe^{3+}$-rich piles with ~18% $(Mg_{0.5}Fe_{0.5})(Si_{0.5}Fe_{0.5})O_3$ Bdg, which is ~1.5% intrinsically denser than the ambient mantle (Fig. 4c), could form large-scale thermochemical structures in the lowermost mantle without being mixed into the background mantle throughout Earth's history (Fig. 5), which may accumulate to form LLSVPs.

Our findings imply that the segregation of $Fe^{3+}$-rich domains may cause heterogeneity in the redox states of the Earth's mantle, that is, the oxygen fugacity of the lowermost mantle may not be as low as inferred in previous studies[54,55]. We also expect these $Fe^{3+}$-rich domains to be enriched in heavy iron isotopes because $Fe^{3+}$ has a larger Fe force constant than $Fe^{2+}$ (ref. [56]), which may affect the iron isotopic features of the deep Earth[57]. The presence of lower-mantle oxidizing heterogeneities would have profound effects on the cycles of volatiles[30] and the deep reservoirs of redox-sensitive elements. For instance, the dense reduced Fe-C/H/S melts formed at the mantle transition zone and shallow

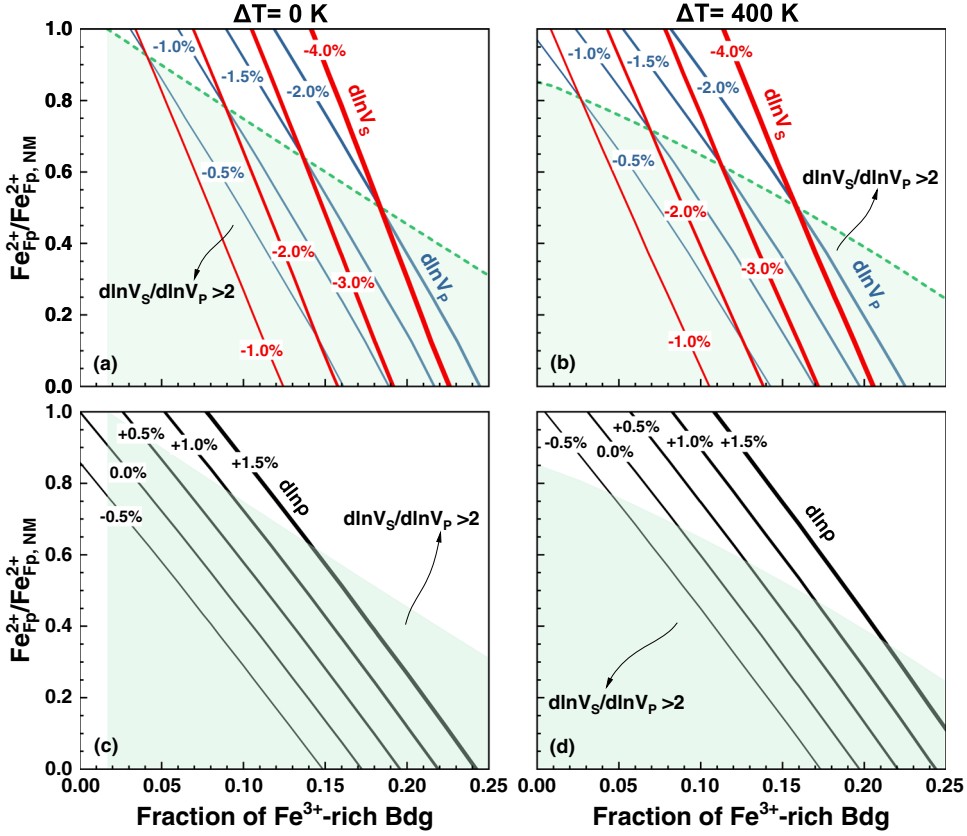

**Fig. 4 Velocity and density anomalies versus $Fe^{2+}$ content and fraction of $(Mg_{0.5}Fe_{0.5})(Si_{0.5}Fe_{0.5})O_3$ bridgmanite. a, b** $V_P$ and $V_S$ anomalies (solid blue and red lines, respectively); **c, d** density perturbation (solid black lines). The line thickness refers to the magnitude as noted by the number. **a, c** Temperature anomaly is 0 K ($\Delta T_{LLSVPS} = 0$ K); **b, d** $\Delta T_{LLSVPS} = +400$ K. The pyrolitic lower mantle is composed of 15% $Mg_{0.82}Fe_{0.18}O$ Fp, 78% $Mg_{0.92}Fe_{0.08}SiO_3$ bridgmanite (Bdg), and 7% $CaSiO_3$ CaPv[37]. $Fe^{2+}_{Fp, NM}$ refers to the FeO content of Fp (18 mol%) in the pyrolite model for the normal lower mantle, and the Fe–Mg partition coefficient between Fp and Bdg[73] is used to constrain their $Fe^{2+}$ contents. The modeled assemblage with different $Fe^{2+}_{Fp}/Fe^{2+}_{Fp, NM}$ ratios has identical mineral fractions to pyrolitic composition in which a portion of $Fe^{2+}$-bearing Bdg was substituted by $(Mg_{0.5}Fe_{0.5})(Si_{0.5}Fe_{0.5})O_3$ Bdg. Dash green lines represent the $dlnV_S/dlnV_P$ ratio of 2.0 and the light green shadows refer to compositional spaces that can reproduce the large $dlnV_S/dlnV_P$ ratio >2.0.

lower mantle by slab/mantle interaction[58,59], if they reach the lowermost mantle, could be converted back to an oxidized state instead of sinking into the core. Dynamic cycling with respect to mantle redox heterogeneity could provide new insights into the thermochemical evolution of the bulk silicate Earth and possibly the oxidation of the atmosphere.

## Methods

**High pressure–temperature experiments.** The experiments were conducted using the 1000-ton multi-anvil apparatus at the University of Michigan. The COMPRES 8/3 and 10/5 cell assemblies were employed in the experiments. The starting material was a mixture of high purity (>99.99%) MgO, $SiO_2$, and $Fe_2O_3$ at a molar ratio of 9:9:1, which corresponds to a bulk composition of $(Mg_{0.9}Fe_{0.1})(Si_{0.9}Fe_{0.1})O_3$. The mixture was heated at 1073 K overnight to remove the moisture and structural water before loading into a platinum capsule. The sample was compressed to target pressure and equilibrated at high temperature for 6–10 h to allow sufficient equilibrium. It was then quenched to room temperature and decompressed to 1 bar.

The recovered sample was polished, coated with carbon, and examined for texture and composition using the JOEL-7800FLV Scanning Electron Microprobe (SEM) and SX-100 Electron Microprobe Analysis (EPMA) at the Electron Microbeam Analysis Laboratory (EMAL) of the University of Michigan. An accelerating voltage of 15 kV and a beam current of 10 nA were employed for imaging and analysis. Forsterite and magnetite were used as standards for Mg, Si, and Fe quantification with EPMA.

**First-principles calculations.** Isothermal elastic tensors ($C^T_{ijkl}$) of crystals in a Cartesian coordinate system usually can be calculated from Eq. (1) (ref. [60]):

$$C^T_{ijkl} = \frac{1}{V}\left(\frac{\partial^2 F}{\partial e_{ij}\partial e_{kl}}\right) + \frac{1}{2}P(2\delta_{ij}\delta_{kl} - \delta_{il}\delta_{kj} - \delta_{ik}\delta_{jl}) \quad (1)$$

where $e_{ij}(i,j = 1,3)$ are infinitesimal strains, $P$ is the isotropic pressure, and $F$ is the Helmholtz free energy, which can be expressed in the quasi-harmonic approximation (QHA) as:

$$F\left(e_{ij}, V, T\right) = U\left(e_{ij}, V\right) + \frac{1}{2}\sum_{q,m}\hbar\omega_{q,m}\left(e_{ij}, V\right) + k_B T\sum_{q,m}ln(1 - \exp(-\frac{\hbar\omega_{q,m}\left(e_{ij}, V\right)}{k_B T}))$$
$$(2)$$

where $V$ is the equilibrium volume of the crystal and $T$ is temperature. Subscripts $q$ and $m$ refer to the phonon wave vector and the normal mode index, respectively. $\hbar$ and $k_B$ are Planck and Boltzmann constants, and $\omega_{q,m}$ is the vibrational frequency of the $i$th mode along with the wave vector $q$. $U$ is the static energy at the equilibrium volume $V$. The second and third terms are the zero-point and vibrational energy contributions, respectively. Adiabatic elastic constants ($C^S_{ijkl}$) can be derived from:

$$C^S_{ijkl} = C^T_{ijkl} + \frac{T}{VC_V}\frac{\partial S}{\partial e_{ij}}\frac{\partial S}{\partial e_{kl}}\delta_{ij}\delta_{kl} \quad (3)$$

where $S$ is the entropy and $C_V$ is the constant volume heat capacity. Therefore, calculations of elastic tensors at high pressure and temperature using this usual method require a vibrational density of states (VDoS) of many strained configurations, which demand a tremendous amount of computational power to calculate based on the DFT. Wu and Wentzcovitch[61] proposed an analytical approach to calculate the thermal contribution to the elastic tensor, only requiring VDoS for unstrained configurations at different equilibrium volumes. This approach greatly reduces the computation cost to the level of <10% of the usual method without loss of accuracy.

To obtain elastic tensors at static conditions and VDoS for unstrained configurations at different equilibrium volumes, we performed first-principles calculations using Quantum Espresso package[62] based on the DFT, plane wave, and pseudopotential. Local density approximation (LDA) was adopted for the exchange-correlation function. The energy cutoff for electronic wave functions was

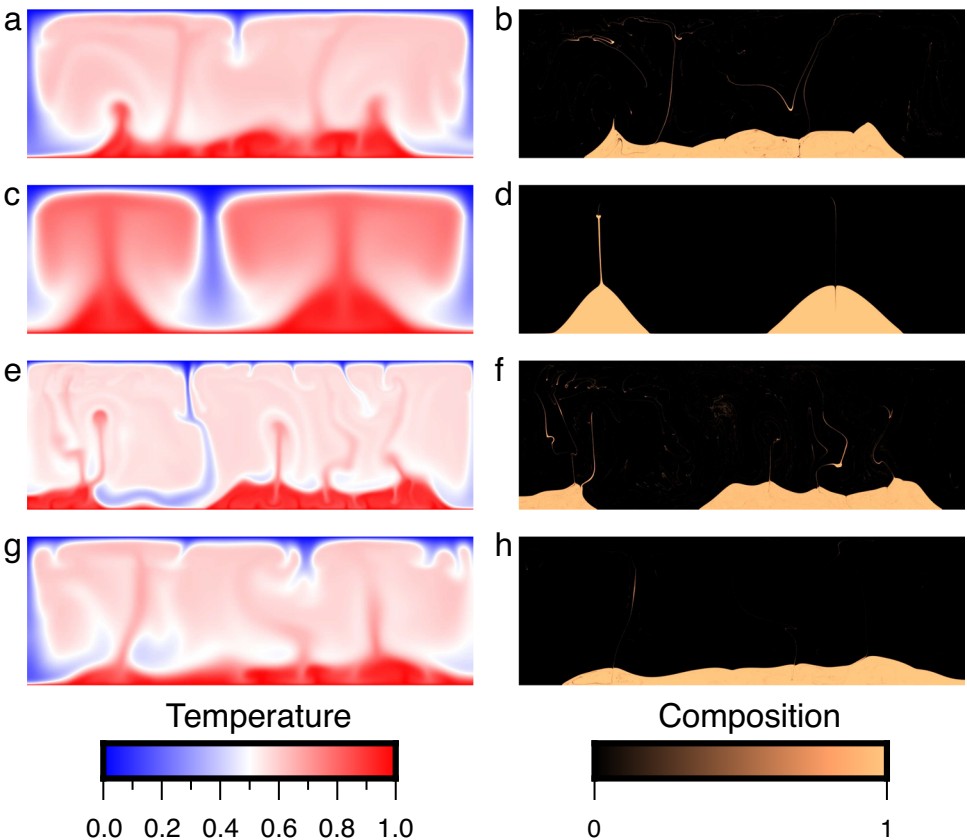

**Fig. 5 Formation of thermochemical piles in the lowermost mantle from iron-rich materials.** The temperature (left column) and composition (right column) fields from the surface to the core–mantle boundary are shown at 4.5 Gyr for case 1 (**a**, **b**), case 2 (**c**, **d**), case 3 (**e**, **f**), and case 4 (**g**, **h**). In cases 1, 2, 3, and 4, the Rayleigh number is $Ra = 1 \times 10^7$, $1 \times 10^6$, $1 \times 10^8$, and $1 \times 10^7$, respectively, and the iron-rich materials (shown by golden colors in the right panels) are 1.2%, 1.2%, 1.2%, and 1.5% intrinsically denser than the background mantle materials (shown by black colors in the right panels), respectively. Large thermochemical piles form after 4.5 Gyr for all cases.

set as 70 Ry. The Mg pseudopotential was generated using the von Barth and Car method for all channels using a 2.5 Bohr cutoff radius and five configurations, $3s^2 3p^0$, $3s^1 3p^1$, $3s^1 3p^{0.5} 3d^{0.5}$, $3s^1 3p^{0.5}$, $3s^1 3d^1$, with weights of 1.5, 0.6, 0.3, 0.3, 0.2, respectively. The pseudopotentials for Si and O were generated using the Troullier-Martins method[63] with the cutoff radius of 1.47 Bohr for Si and 1.45 Bohr for O. Valence configurations for Si and O are $3s^2 3p^4 3d^0$ and $2s^2 2p^4$, respectively. The pseudopotentials for Al and Fe were generated using the Vanderbilt method[64] with a valence configuration of $3s^2 3p^1$ and a cutoff radius of 1.77 Bohr for Al, and a valence configuration of $3s^2 3p^6 3d^{6.5} 4s^1 4p^0$ and a cutoff radius of 1.8 Bohr for Fe. To address the large on-site Coulomb interactions among the localized electrons (Fe 3d electrons)[65], we introduced a Hubbard $U$ correction to the LDA (LDA+$U$) for all DFT calculations. $U$ values for $Fe^{3+}$ on A and B sites in bridgmanite were non-empirically determined using linear response method[66] in previous work[43] and adopted in this study. The initial structure was constructed by replacing one nearest-neighbor $Mg^{2+}$–$Si^{4+}$ pair with one $[Fe^{3+}]_{Mg}$–$[Fe^{3+}]^{Si}$ pair[43–45]. Crystal structures at variable pressures were well optimized on a $6 \times 6 \times 4$ k-point mesh, and VDoS were calculated using the finite displacement method as implemented in the code PHONOPY[67]. The elastic tensors at static conditions were calculated from the linear dependence of stress on the small strain. Owing to the enormous computational cost of calculating the vibrational density of state based on LDA+$U$, we used the 20-atom unit cell for $(Mg_{0.5}Fe_{0.5})(Si_{0.5}Fe_{0.5})O_3$ bridgmanite[43–45]. Similar to previous studies on the elastic properties of Fe-bearing bridgmanite[45,46], we only report results for aggregate elastic moduli, not individual elastic coefficients. The latter are sensitive to atomic configurations and therefore to supercell size, which can accommodate different configurations for the same composition. The aggregate elastic moduli, $K_S$ and $G$, are quite insensitive to the atomic configuration[68–70].

The Helmholtz free energy calculated from Eq. (2) within the QHA versus volume was fitted by the isothermal third-order finite strain equation of state, and then we can obtain all thermodynamic properties, such as pressures at different temperatures and volumes. Our results show that $(Mg_{0.5}Fe_{0.5})(Si_{0.5}Fe_{0.5})O_3$ bridgmanite has a larger volume than $(Mg_{0.5}Fe_{0.5})(Si_{0.5}Al_{0.5})O_3$ bridgmanite (Fig. 2). The substitution of $Al^{3+}$ for LS $Fe^{3+}$ in the octahedral site causes a slight decrease of ~1.0% in volume at >80 GPa. Compared to pristine Bdg ($MgSiO_3$), these two $Fe^{3+}$- and $Al^{3+}$-rich species have larger volumes; for example, at 90 GPa

and 2000 K, the volumes of $(Mg_{0.5}Fe_{0.5})(Si_{0.5}Al_{0.5})O_3$ and $(Mg_{0.5}Fe_{0.5})(Si_{0.5}Fe_{0.5})O_3$ are 3.5% and 4.4% larger than that of $MgSiO_3$, respectively.

Using the equation of states, we transferred volume- and temperature-dependent elasticity into pressure- and temperature-dependent elasticity. The adiabatic bulk modulus $K_S$ and shear modulus $G$ can be obtained by computing the Voigt–Reuss–Hill averages[71] from elastic tensors. Thus, compressional and shear velocities can be calculated from the equations $V_P = \sqrt{(K_S + \frac{4}{3}G)/\rho}$ and $V_S = \sqrt{G/\rho}$ ($\rho$ is density). Bulk moduli ($K_S$), shear moduli ($G$), compressional-wave velocity ($V_P$), and shear-wave velocity ($V_S$) are also derived from LDA+$U$ calculations as shown in Supplementary Fig. 3. Compared to $(Mg_{0.5}Fe_{0.5})(Si_{0.5}Fe_{0.5})O_3$ Bdg, $(Mg_{0.5}Fe_{0.5})(Si_{0.5}Al_{0.5})O_3$ Bdg has a lower density, similar $K_S$ but much larger $G$ at >90 GPa, which results in much higher velocities in $(Mg_{0.5}Fe_{0.5})(Si_{0.5}Al_{0.5})O_3$ Bdg. At 100 GPa and 2000 K, the differences in density, $K_S$, $G$, $V_P$, and $V_S$ between $(Mg_{0.5}Fe_{0.5})(Si_{0.5}Al_{0.5})O_3$ and $(Mg_{0.5}Fe_{0.5})(Si_{0.5}Fe_{0.5})O_3$ are −10.4%, −0.9%, 19.2%, 8.1%, and 14.8%, respectively. Elastic moduli and velocities almost linearly depend on pressure and temperature after B-site $Fe^{3+}$ spin transition and their first pressure and temperature derivatives are comparable to those of $MgSiO_3$ Bdg (Supplementary Table 3).

**Ab initio investigation on the stability of $(Mg_{0.5}Fe_{0.5})(Si_{0.5}Fe_{0.5})O_3$ bridg-manite.** Through high-pressure and high-temperature experiments, we find the formation of iron-rich bridgmanite $(Mg_{0.5}Fe_{0.5})(Si_{0.5}Fe_{0.5})O_3$ coexisting with Fe-poor bridgmanite, instead of forming a single phase of $(Mg_{0.9}Fe_{0.1})(Si_{0.9}Fe_{0.1})O_3$ Bdg with homogeneous iron content. In order to check the relative stability of $(Mg_{0.5}Fe_{0.5})(Si_{0.5}Fe_{0.5})O_3$ bridgmanite, we also calculated the formation energy of the decomposition reaction:

$$(Mg_{0.875}Fe_{0.125})(Si_{0.875}Fe_{0.125})O_3 \leftrightarrow 3/4MgSiO_3 + 1/4(Mg_{0.5}Fe_{0.5})(Si_{0.5}Fe_{0.5}) \quad (4)$$

The Gibbs free energy of $(Mg_{1-x}Fe_x)(Si_{1-x}Fe_x)O_3$ can be expressed as (see Supplementary Materials):

$$G_{HS/LS}(P, T) = G_{HS/LS}^{stat+vib}(P, T) + G_{HS/LS}^{mag}(P, T) - TS^{conf} \quad (5)$$

where $S^{conf}$ is the configurational entropy ($S^{conf} = k_B \ln M$, $M$ is the configuration degeneracy). $G_{HS/LS}^{stat+vib}(P, T)$ and $G_{HS/LS}^{mag}(P, T)$ can be derived from

Eqs. (3–6) in Supplementary Materials. Thus, the Gibbs formation free energy of the decomposition reaction for pure HS/LS state can be expressed as:

$$\Delta G = \frac{1}{4} * (G^{\text{stat+vib}}_{x=0.5} - TS^{\text{conf}}_{x=0.5}) + \frac{3}{4} * G^{\text{stat+vib}}_{\text{MgSiO3}} - (G^{\text{stat+vib}}_{x=0.125} - TS^{\text{conf}}_{x=0.125}) \quad (6)$$

We also investigated the disordered substitution of $Fe^{3+}$ in $(Mg_{0.875}Fe_{0.125})$ $(Si_{0.875}Fe_{0.125})O_3$ bridgmanite in a 40-atom cell ($\sqrt{2} \times \sqrt{2} \times 1$ supercell). Similar to the case for $(Mg_{0.5}Fe_{0.5})(Si_{0.5}Fe_{0.5})O_3$ bridgmanite, the initial structure was constructed by replacing one nearest-neighbor $Mg^{2+}-Si^{4+}$ pair with one $[Fe^{3+}]_{Mg}-[Fe^{3+}]^{Si}$ pair. Due to the extremely high computational cost of VDOS calculation using LDA+$U$ functional, we did not calculate the VDOS of $(Mg_{0.875}Fe_{0.125})$ $(Si_{0.875}Fe_{0.125})O_3$ bridgmanite. Because $(Mg_{1-x}Fe_x)(Si_{1-x}Fe_x)O_3$ bridgmanite with different $Fe^{3+}$ contents have similar structures, here we assume that the vibrational contribution to the Gibbs free energy linearly depends on $Fe^{3+}$ content ($G^{\text{vib}}_{x=0.125} = \frac{1}{4} * G^{\text{vib}}_{x=0.5} + \frac{3}{4} * G^{\text{vib}}_{\text{MgSiO3}}$). Under this approximation, $\Delta G$ can be written as:

$$\Delta G = \frac{1}{4} * (H^{\text{stat}}_{x=0.5} - TS^{\text{conf}}_{x=0.5}) + \frac{3}{4} * H^{\text{stat}}_{\text{MgSiO3}} - (H^{\text{stat}}_{x=0.125} - TS^{\text{conf}}_{x=0.125}) \quad (7)$$

where $H^{\text{stat}}$ is the internal energy or the Gibbs free energy without the vibrational contribution.

As shown in Supplementary Fig. 2, $\Delta G$ is negative at lower-mantle conditions regardless of the spin state of B-site $Fe^{3+}$ in $(Mg_{1-x}Fe_x)(Si_{1-x}Fe_x)O_3$ bridgmanite. This implies that the assemblage of $(Mg_{0.5}Fe_{0.5})(Si_{0.5}Fe_{0.5})O_3$ and MgSiO3 bridgmanite is more stable than the single-phase $(Mg_{0.875}Fe_{0.125})(Si_{0.875}Fe_{0.125})O_3$ bridgmanite, consistent with our experimental results (Fig. 1). In addition, our LDA+$U$ calculations extend the occurrence of this decomposition reaction to the lowermost-mantle conditions, where $(Mg_{0.5}Fe_{0.5})(Si_{0.5}Fe_{0.5})O_3$ bridgmanite is still more stable than $(Mg_{0.875}Fe_{0.125})(Si_{0.875}Fe_{0.125})O_3$ bridgmanite.

In order to check the effect of the exchange-correlation function on the results, we also calculated the enthalpy change of this reaction using the generalized gradient approximation with Hubbard $U$ correction (GGA+$U$). $U$ values for $Fe^{3+}$ on A and B sites in bridgmanite are 3.3 and 4.5 eV, respectively. The $\Delta H$ predicted by the GGA+$U$ calculations is similar to that from the LDA+$U$ calculations (Supplementary Fig. 2), both of which favor the mixed phases over the single phase.

**Thermoelastic models for the estimations of velocity and density heterogeneities**. Previous studies[37] have suggested that the pyrolitic lower mantle that consists of 15% $Mg_{0.82}Fe_{0.18}O$ ferropericlase (Fp), 78% $Mg_{0.92}Fe_{0.08}SiO_3$ bridgmanite ($Fe^{2+}$-Bdg), and 7% CaSiO3 Ca-perovskite (CaPv)) can admirably reproduce the velocity and density profiles of PREM model[72] for the lower mantle. Combining the thermoelastic properties[20,46,47] of these three major minerals with our elastic data for LS-$(Mg_{0.5}Fe_{0.5})(Si_{0.5}Fe_{0.5})O_3$ and $(Mg_{0.5}Fe_{0.5})(Si_{0.5}Al_{0.5})O_3$ bridgmanite, we quantify the dependences of velocity and density anomalies on the amount of $Fe^{3+}$-rich Bdg by substituting a certain proportion of $Fe^{3+}$-free Bdg with $(Mg_{0.5}Fe_{0.5})(Si_{0.5}Fe_{0.5})O_3$ bridgmanite. Compared to the pyrolitic composition, the modeling chemical assemblage has identical mineral fractions (15% ferropericlase + 78% Bdg + 7% CaPv) in which a portion of $Fe^{2+}$-bearing Bdg was substituted by $(Mg_{0.5}Fe_{0.5})(Si_{0.5}Fe_{0.5})O_3$ or $(Mg_{0.5}Fe_{0.5})(Si_{0.5}Fe_{0.4}Al_{0.1})O_3$ Bdg. In other words, the ferropericlase and Ca-perovskite contents are fixed to 15% and 7%, respectively. We also explore the effect of $Fe^{2+}$ content in the assemblage (noted by the $Fe^{2+}$ content in Fp, $Fe^{2+}_{Fp}$) on modeling results because the incorporation of $Fe^{2+}$ into Fp and Bdg decreases their velocities to some extent[46,47]. The Fe–Mg partition coefficient between Fp and Bdg[73] is used to constrain their $Fe^{2+}$ contents. The modeled aggregate with enrichment of $Fe^{3+}$-rich Bdg is composed of 15% $Mg_{1-x}Fe_xO$ ferropericlase, 7% CaSiO3 Ca-perovskite, Z% $(Mg_{0.5}Fe_{0.5})(Si_{0.5}Fe_{0.5})O_3$ or $(Mg_{0.5}Fe_{0.5})(Si_{0.5}Fe_{0.4}Al_{0.1})O_3$ bridgmanite, and (78-Z)% $Mg_{1-y}Fe_ySiO_3$ bridgmanite.

The elastic moduli and densities of the aggregate are calculated using:

$$\rho = \sum_i f_i \rho_i \quad (8)$$

$$M = \frac{1}{2} \left[ \sum_i f_i M_i + \left( \sum_i f_i M_i^{-1} \right)^{-1} \right] \quad (9)$$

where $\rho_i$, $M_i$, and $f_i$ are the density, bulk modulus ($K_S$) or shear modulus ($G$), and the fraction of the $i$th mineral, respectively. Similarly, the compressional and shear velocities ($V_P$ and $V_S$) were derived from $V_P = \sqrt{(K_S + \frac{4}{3}G)/\rho}$ and $V_S = \sqrt{G/\rho}$, and hence the velocity and density anomalies relative to the pyrolitic lower mantle are estimated with the consideration of temperature anomaly.

**Geodynamic models**. We performed thermochemical calculations to study the dynamics of iron-rich materials in the lowermost mantle. The numerical simulations were conducted by solving the nondimensional equations of conservation of mass, momentum, and energy under the Boussinesq approximation. The intrinsic density anomaly is represented by the buoyancy number $B$, which is defined as the ratio between the intrinsic (compositional) density anomaly and the density anomaly caused by thermal expansion, $B = \Delta\rho/(\rho\alpha\Delta T)$, where $\Delta\rho$ is the intrinsic density anomaly for the iron-rich materials compared to the background mantle. We used $\alpha = 1 \times 10^{-5} \text{K}^{-1}$ and $\Delta T = 2500 \text{K}$ in this study.

The whole-mantle dynamics is simulated in a 2D Cartesian box with an aspect ratio of 3:1. The model domain contains 1536 and 512 elements in the horizontal and vertical directions, respectively. Models are basally heated, with the top and bottom having a fixed temperature at $T = 0$ and $T = 1$, respectively. The top and bottom boundaries are both free-flip and the side boundaries are reflective. The viscosity is determined by $\eta = \eta_0 \exp[0.5 - T]$, where $\eta_0$ is a viscosity pre-factor, and $A$ is the activation energy. Here, $\eta_0$ is 1.0 and 30.0 for the upper mantle and the lower mantle, respectively, and $A = 9.21$ which leads to the viscosity changing four orders of magnitude as temperature increases from 0 to 1. Initially, the whole mantle was assumed to be hot with a temperature of $T = 0.72$ (or 1800 K) everywhere, and we introduced a global layer of iron-rich materials in the lowermost 300 km of the mantle.

We performed four cases. Case 1 is the reference case in which the iron-rich materials are 1.2% intrinsically denser (with a buoyancy number of $B = 0.48$) than the background mantle materials and the Rayleigh number, which controls the vigor of mantle convection, is $Ra = 1 \times 10^7$. Cases 2 and 3 have a Rayleigh number of $Ra = 1 \times 10^6$ and $Ra = 1 \times 10^8$, respectively, while other parameters are the same as case 1. In case 4, the iron-rich materials are 1.5% intrinsically denser (with a buoyancy number of $B = 0.6$) than the background mantle materials, and other parameters are the same as case 1.

## Data availability
The data are available in the main text, the supplementary materials, and from the corresponding authors.

## Code availability
The open-source Quantum Espresso package used in this study is available at https://www.quantum-espresso.org/.

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

## Acknowledgements

This work is supported by the Strategic Priority Research Program (B) of the Chinese Academy of Sciences (XDB41000000 and XDB18000000), Natural Science Foundation of China (41925017 and 41721002), and the Fundamental Research Funds for the Central Universities (WK2080000144). M.M.L. is supported by National Science Foundation (NSF) grants EAR-1849949 and EAR-1855624. S.M.D. acknowledges support from the new faculty startup funding of Michigan State University and NSF EAR-1664332. J. Li acknowledges support from NSF EAR2031149 and NASA NNX15AG54G. Some computations were conducted in the Supercomputing Center of the University of Science and Technology of China.

## Author contributions

W.Z.W. and J.C.L. conceived and designed this project. W.Z.W. performed the theoretical calculations. J.C.L., F.Z., S.M.D., and J.L. performed the experiments, M.M.L. conducted the geodynamic simulations. W.Z.W., J.C.L., and Z.F. wrote the paper, and all authors contributed to the discussion of the results and revision of the paper.

## Competing interests

The authors declare no competing interests.
