## [Peer Review File · Nature Communications]

REVIEWER COMMENTS

Reviewer #1 (Remarks to the Author):

The paper by Wang et al. presents one possible chemical origin of large low shear velocity provinces (LLSVPs) based on their results using experimental, computational and geo-dynamic modeling approaches. This region is so complex that its origin can be chemical or thermal or dynamical, needing more studies.

This paper proposes an explanation, which appears to be quite appealing. It is related to iron-bearing bridgmanite, which is a major material of Earth's lower mantle. Analyses and discussion of the results are done very thoroughly to support the idea that iron-rich bridgmanite, $(\text{Mg}_{0.5}\text{Fe}_{0.5})(\text{Si}_{0.5}\text{Fe}_{0.5})\text{O}_3$ may be responsible for LLSVPs. This perhaps makes sense because iron-rich bridgmanite tends to have lower elastic wave velocities and higher density than co-existing Fe-poor bridgmanite. I believe that the results and conclusions provide further insight into the origin of LLSVPs.

Here are a few suggestions/comments for the authors to consider:

a) The authors evaluated Gibbs energy of $(\text{Mg}_{0.5}\text{Fe}_{0.5})(\text{Si}_{0.5}\text{Fe}_{0.5})\text{O}_3$ and MgSiO_3 relative to $(\text{Mg}_{0.875}\text{Fe}_{0.125})(\text{Si}_{0.875}\text{Fe}_{0.125})\text{O}_3$ to show the mixture is energetically more favorable. Their evaluation of Gibbs energy used quasi-harmonic approximation and entropy terms. It would be helpful if the enthalpy for this reaction was also presented for each case of spin state to see whether enthalpy comparisons also favor the mixed phase over the single phase.

b) It would be helpful to show actual simulation results in addition to model-based curves. For instance, Figure 2 should show the calculated pressure-volume values along 300 K curve or may be static (0 K) results.

c) The authors used LDA+U as opposed to GGA+U. Generally, GGA is considered to be more appropriate to deal with transition metal elements such as iron. Maybe they should do some GGA+U calculations to assess the effects of LDA versus GGA.

d) Also, it would be interesting to see whether just LDA calculations will lead to the same findings and conclusions. This may be the case because they appear to be insensitive to the spin state.

e) While the paper reads well, there appear to be several typos, for instance,

line 190: required
line 204: anomalies
line 206: fraction

...

Reviewer #2 (Remarks to the Author):

The paper describes calculations to determine the stability and elastic properties of the lower mantle mineral bridgmanite with a composition that lies mid-way between the end members MgSiO_3 and FeFeO_3 i.e. where 50 % of both bridgmanite cation sites are occupied by ferric iron. The authors argue that a miscibility gap exists between high and low ferric iron bearing

bridgmanite such that a mixture of the two phases is more stable than an intermediate solid solution. Using ab initio calculations they show that the shear modulus of the high ferric iron bearing bridgmanite is very low compared to the MgSiO₃ end member whereas the bulk modulus is not so low. They argue that mantle containing this ferric iron enriched bridgmanite would therefore have seismic properties that could match those observed for large low shear velocity provinces (LLSVP). While the results are interesting there is insufficient data through which to evaluate the ab initio calculations. Furthermore the scenario in which such a high ferric iron bearing region of the mantle could form is not at all clear and does not seem to be logical when considered in a broader light. The properties of LLSVPs are quite vague and without a reasonable argument as to why such a region should form with the proposed composition, it is hard to accept the scenario based on the broad similarity with seismic observations alone. For these reasons it is hard to recommend publication.

Detailed comments

1) The main problem with the author's scenario is why would such an enormously ferric iron enriched region of the mantle ever form? If we assume that the authors are correct and there is a ferric iron enriched form of bridgmanite that coexists on the other side of a miscibility gap from MgSiO₃ bridgmanite, how did the mantle ever get so enriched in ferric iron such that the bulk composition can enter this gap? The main reason people have argued that high ferric iron contents can be obtained in the lower mantle is through charge disproportionation of Fe²⁺ to create Fe³⁺ and iron metal. The problem with this scenario is that while it raises the ferric/ferrous ratio, in the end the total iron content of silicates decreases because iron metal is formed. This process can turn a significant proportion of ferrous iron in to ferric iron but it cannot raise the ferric iron component of the mantle independently. Also disproportionation, if it occurs, is driven by the requirement of a particular phase for a raised level of ferric iron, while in equilibrium with iron metal. Once this is satisfied for normal MgSiO₃ bridgmanite, why should more ferric iron be produced to stabilise an additional ferric iron enriched bridgmanite? What process would enrich the mantle in ferric iron to a level which is in excess of what the existing mantle minerals can accommodate?

2) The authors also mention subduction of banded iron formations which are certainly ferric iron enriched but there are also problems with this scenario. Putting aside for the moment arguments as to whether they would indeed have formed in deep sea environments or whether such accumulations would be subducted or accreted to continents, comes the issue that the sheer mass of material required i.e. 3-9 % of the entire mantle, would be far outside of estimates for how much of these formations we would expect to find. It must be born in mind that banded iron formations are produced as insoluble ferric iron is formed through reaction with oxygen produced by photo synthesis. Therefore to balance the ferric iron produced there needs to be a complementary reservoir of reduced carbon that would need to be very very significant.

3) The authors also do not consider that MgSiO₃ bridgmanite can already accommodate a significant proportion of ferric iron as demonstrated in a number of studies such as those of Catalli et al. (2010) and Hummer and Fei (2012). The results of the current study are actually not discussed in light of these previous studies.

4) The most important results from this study are those of the elastic properties of the ferric iron enriched phase. However apart from a couple of diagrams we get very few details on these properties. What are the individual elastic constants for example? What are the room pressure and temperature moduli? There is no evaluation of these results either. Surely with an ab initio calculation of the structure it should be possible to assess why the shear modulus is affected so strongly by the coupled substitution of ferric iron. Also the effect on the bulk modulus of ferric iron coupled substitution could be compared with the results of Catalli et al. (2010).

5) There are only very vague comparisons made with the expected properties of LLSVPs. The expected range of LLSVP properties should be plotted in figures 3 and 4. What is a suitable criterion for a good agreement? Some assessment of this would be required. LLSVPs do not show up significantly in terms of the bulk sound velocity which provides a further criterion.

Minor comments

Lines 165 and 168 there is no explanation for the difference between these two assessments- exactly the same comparison is made with two different answers.

The y axis in figure 4 is very difficult to understand. Why is the difference between Fe₂+Fp and Fe₂+Fp,pyrolite?

Reviewer #3 (Remarks to the Author):

Dear editor, dear authors,

The paper by Wang et al. attempts to unravel the origin of LLSPVs by combining high-pressure experiments with ab-initio calculations and geodynamic modelling. Understanding the origin of LLSPVs is a pivotal question in geophysics and the present study is thus very timely; it will certainly be of interest to the community.

The present manuscript proposes that iron-rich bridgmanite forms in the lower mantle and causes the notable seismic features of large low shear velocity provinces. Iron enrichment generally leads to lower seismic wave velocities and an increase in density for mantle and it is thus does not seem surprising that a model containing iron-rich bridgmanite (when mixed with variable amounts of "normal" mantle) can explain the seismic features of LLSPVs. The novelty of this paper is to show that iron-rich bridgmanite might indeed exist in the lower mantle. To reach this conclusion the paper relies on experimental work as well as ab-initio calculations. My major concerns are the following:

1. The authors report results from a single multi-anvil experiment at lower mantle conditions (at 24GPa, 1873K). In this experiment, they observed co-existing Fe-poor bridgmanite and Fe-rich akimotoite. They conclude that the observed akimotoite back-transformed from bridgmanite. But is there really a stable assemblage of Fe-rich and Fe-poor bridgmanite during the experiment (and in the lower mantle) or is the observed exsolution a result of the experimental procedure, i.e. a result of experimental conditions too close to the akimotoite stability field and/or a quenching result? I feel that more experiments at lower mantle conditions would be helpful to reach a robust conclusion. Also, how does the results depend on pressure. Will the exsolution process happen in the deeper mantle as well? In-situ diamond-anvil cell experiments could be helpful since they would allow for a direct observation of stable phase assemblages at deep lower mantle conditions. I note that the authors have performed ab-initio calculations to confirm their experimental findings and show that the Gibbs Free Energy of an assemblage of Fe-rich/Fe-poor bridgmanite is energetically favorable as opposed to a solid solution. However, the experimental verification seems weak to me.
2. Both experiments and calculations are reported for a pure MgSiO₃-Fe₂O₃ system. How does the system change when considering a bulk lower mantle composition, i.e. does the presence of ferropiclasite and CaPv perovskite change the stable phase assemblage? Also, how does Al-incorporation (that is expected in the lower mantle) change the equilibrium phase assemblage?
3. According to the reported findings, about 15% of (Mg_{0.5}Fe_{0.5})(Si_{0.5}Fe_{0.5})O₃ bridgmanite is required to explain the seismic features of the LLSPVs. This sounds quite a lot to me. Can they authors provide some rough calculations to show that the required Fe³⁺ could indeed originate from the disproportionation of Fe²⁺ in an early magma ocean as suggested in the manuscript (motivated by recent high-pressure experiments, again at shallow lower mantle conditions (Armstrong et al. 2019))?

Further points to consider:

- The authors have done calculations of the elastic constants (C_{ij}) of Fe-bridgmanite and Fe/Al-bridgmanite. The results are of general interest to the geophysics community and should be tabulated along with the bulk elastic properties. It would also be nice to see more careful

comparison of the results with previous calculations and available experimental measurements of wave velocities (acknowledging limitations in the compositional and pressure/temperature ranges that have been studied), some that come to mind include Kurnosov et al. 2017, Fu et al. 2018, Fu et al. 2019.

- There are some typos throughout the manuscript, e.g. "origin" in line 45, "reported" in line 91, "corresponding" in line 107, "bulk modulus" in line 157,.. Please check and correct.

In summary, the paper presents interesting findings with possibly important consequences. Unfortunately, the conclusions fundamentally rely on the assumption that iron-rich bridgmanite forms as stable phase in the lower mantle, and that it exists a sufficiently large amount. At this point, I am not convinced that the experimental results presented in the paper provide robust evidence for this process to happen in the lower mantle. Having said that, it is worth re-iterating the results are of significant interest and I am convinced that they will trigger follow-up research to test the here-hypothesized nature of LLSVPs.

I hope that my comments can help the authors to improve the quality of the manuscript and can assist the editor to reach a balanced decision.

Armstrong, K., D. J. Frost, C. A. McCammon, D. C. Rubie and T. Boffa Ballaran (2019). Deep magma ocean formation set the oxidation state of Earth's mantle. *Science* 365(6456): 903-906.

Fu, S., J. Yang, N. Tsujino, T. Okuchi, N. Purevjav and J.-F. Lin (2019). Single-crystal elasticity of (Al,Fe)-bearing bridgmanite and seismic shear wave radial anisotropy at the topmost lower mantle. *Earth and Planetary Science Letters* 518: 116-126.

Fu, S., J. Yang, Y. Zhang, T. Okuchi, C. McCammon, H.-I. Kim, S. K. Lee and J.-F. Lin (2018). Abnormal Elasticity of Fe-Bearing Bridgmanite in the Earth's Lower Mantle. *Geophysical Research Letters* 45(10): 4725-4732.

Kurnosov, A., H. Marquardt, D. J. Frost, T. B. Ballaran and L. Ziberna (2017). Evidence for a Fe³⁺-rich pyrolitic lower mantle from (Al,Fe)-bearing bridgmanite elasticity data. *Nature* 543(7646): 543-546.

(Our replies are in blue and the revisions in the manuscript are red)

Reviewer #1

Comment 1

The paper by Wang et al. presents one possible chemical origin of large low shear velocity provinces (LLSVPs) based on their results using experimental, computational and geo-dynamic modeling approaches. This region is so complex that its origin can be chemical or thermal or dynamical, needing more studies.

This paper proposes an explanation, which appears to be quite appealing. It is related to iron-bearing bridgmanite, which is a major material of Earth's lower mantle. Analyses and discussion of the results are done very thoroughly to support the idea that iron-rich bridgmanite, $(\text{Mg}_{0.5}\text{Fe}_{0.5})(\text{Si}_{0.5}\text{Fe}_{0.5})\text{O}_3$ may be responsible for LLSVPs. This perhaps makes sense because iron-rich bridgmanite tends to have lower elastic wave velocities and higher density than co-existing Fe-poor bridgmanite. I believe that the results and conclusions provide further insight into the origin of LLSVPs.

Reply: Thanks for recognizing our contribution.

Comment 2

The authors evaluated Gibbs energy of $(\text{Mg}_{0.5}\text{Fe}_{0.5})(\text{Si}_{0.5}\text{Fe}_{0.5})\text{O}_3$ and MgSiO_3 relative to $(\text{Mg}_{0.875}\text{Fe}_{0.125})(\text{Si}_{0.875}\text{Fe}_{0.125})\text{O}_3$ to show the mixture is energetically more favorable. Their evaluation of Gibbs energy used quasi-harmonic approximation and entropy terms. It would be helpful if the enthalpy for this reaction was also presented for each case of spin state to see whether enthalpy comparisons also favor the mixed phase over the single phase.

Reply: We calculated the enthalpy of formation for the decomposition of $(\text{Mg}_{0.875}\text{Fe}_{0.125})(\text{Si}_{0.875}\text{Fe}_{0.125})\text{O}_3$ into $(\text{Mg}_{0.5}\text{Fe}_{0.5})(\text{Si}_{0.5}\text{Fe}_{0.5})\text{O}_3$ and MgSiO_3 bridgmanite. As shown in Supplementary Fig. 2, the change of enthalpy also favors the mixed phases over the single phase.

Comment 3

It would be helpful to show actual simulation results in addition to model-based curves. For instance, Figure 2 should show the calculated pressure-volume values along 300 K curve or may be static (0 K) results.

Reply:

The calculation details of equation of state and elasticity can be found in **Method and Supplementary Information**. The curves shown in Fig. 2 are the calculated unit-cell volumes of $(\text{Mg}_{0.5}\text{Fe}_{0.5})(\text{Si}_{0.5}\text{Fe}_{0.5})\text{O}_3$ and $(\text{Mg}_{0.5}\text{Fe}_{0.5})(\text{Si}_{0.5}\text{Al}_{0.5})\text{O}_3$ bridgmanite at 20-140 GPa and 300-3000 K.

We updated the figure caption for clarification in line 649-655: “**Figure 2. Isothermal compression curves predicted *ab initio* calculations for $(\text{Mg}_{0.5}\text{Fe}_{0.5})(\text{Si}_{0.5}\text{Fe}_{0.5})\text{O}_3$ and $(\text{Mg}_{0.5}\text{Fe}_{0.5})(\text{Si}_{0.5}\text{Al}_{0.5})\text{O}_3$ bridgmanite. The blue, green, orange, and red curves are calculated compression curves of $(\text{Mg}_{0.5}\text{Fe}_{0.5})(\text{Si}_{0.5}\text{Fe}_{0.5})\text{O}_3$ and $(\text{Mg}_{0.5}\text{Fe}_{0.5})(\text{Si}_{0.5}\text{Al}_{0.5})\text{O}_3$ bridgmanite at 300 K, 1000 K,**

2000 K and 3000 K, respectively. The blue squares are experimental measurements from Liu et al. (2018), which shows that the spin transition of Fe^{3+} in the B-site of $(\text{Mg}_{0.5}\text{Fe}_{0.5})(\text{Si}_{0.5}\text{Fe}_{0.5})\text{O}_3$ bridgmanite occurs between 43-53 GPa at 300 K. The blue circles are experimental results of Zhu et al. (2020).”

Comment 4

The authors used LDA+U as opposed to GGA+U. Generally, GGA is considered to more appropriate to deal with transition metal elements such as iron. Maybe they should do some GGA+U calculations to assess the effects of LDA versus GGA.

Also, it would be interesting to see whether just LDA calculations will lead to the same findings and conclusions. This may be the case because they appear to be insensitive to the spin state.

Reply:

We used GGA+U to investigate the Gibbs formation free energy of the decomposition of $(\text{Mg}_{0.875}\text{Fe}_{0.125})(\text{Si}_{0.875}\text{Fe}_{0.125})\text{O}_3$ into $(\text{Mg}_{0.5}\text{Fe}_{0.5})(\text{Si}_{0.5}\text{Fe}_{0.5})\text{O}_3$ and MgSiO_3 bridgmanite. As shown in Supplementary Fig. 2, the complementary calculations based on GGA+U also support our conclusion that the mixed phases are more stable than the single phase. Please refer to the discussion in lines 599-604.

The approximated functionals based on both LDA and GGA have been very successful in predicting the properties of many materials. However, they cannot describe the large on-site Coulomb interactions among the localized electrons in open shell configurations sufficiently. Introducing a Hubbard U correction to the LDA and GGA overcomes this limitation, and therefore, we do not use the LDA or GGA to predict the energy.

Comment 5

While the paper reads well, there appear to be several typos, for instance,

line 190: required

line 204: anomalies

line 206: fraction

Reply: Thank you for catching the typos and we corrected them accordingly.

Reviewer #2

Comment 1

The paper describes calculations to determine the stability and elastic properties of the lower mantle mineral bridgmanite with a composition that lies mid-way between the end members MgSiO_3 and FeFeO_3 i.e. where 50 % of both bridgmanite cation sites are occupied by ferric iron. The authors argue that a miscibility gap exists between high and low ferric iron bearing bridgmanite such that a mixture of the two phases is more stable than an intermediate solid solution. Using ab initio calculations they show that the shear modulus of the high ferric iron bearing bridgmanite is very low compared to the MgSiO_3 end member whereas the bulk modulus is not so low. They argue that mantle containing this ferric iron enriched bridgmanite would therefore

have seismic properties that could match those observed for large low shear velocity provinces (LLSVP). While the results are interesting there is insufficient data through which to evaluate the *ab initio* calculations. Furthermore the scenario in which such a high ferric iron bearing region of the mantle could form is not at all clear and does not seem to be logical when considered in a broader light. The properties of LLSVPs are quite vague and without a reasonable argument as to why such a region should form with the proposed composition, it is hard to accept the scenario based on the broad similarity with seismic observations alone. For these reasons it is hard to recommend publication.

Reply:

The details for our *ab initio* calculations have been provided in the method part and all results including the equation of state, elastic properties, and spin transition are included in the main text and supplementary materials. In this work, we only report results for aggregate elastic moduli, not individual elastic coefficients. The latter are sensitive to atomic configurations and therefore to supercell size, which can accommodate different configurations for the same composition. The aggregate elastic moduli, K and G, are quite insensitive to the atomic configuration, which has been verified in previous studies (Nunez-Valdez et al., 2010, 2012a, 2012b, 2013; Marcondes et al., 2015). This strategy was also adopted in previous theoretical studies on the elastic properties of Fe-bearing bridgmanite (Shukla et al., 2015, 2016) with the same method as the one used in this study.

The only precondition needed to produce the ferric-iron-rich bridgmanite is regional enrichment in iron, which is a common scenario invoked to explain LLSVPs and has been hypothesized to be consistent with the solidification of an early magma ocean (more elaboration below). The remnants of a basal magma ocean created early in Earth's history would be enriched in iron (Garnero et al., 2016; Labrosse et al., 2007; Lee et al., 2010; Nomura et al., 2011). Ferrous iron in dense silicate melts would partially disproportionate to Fe^{3+} plus Fe^0 at high pressures (Armstrong et al., 2019), and segregation of precipitated Fe^0 from the magma ocean into core would enrich silicate melt in Fe_2O_3 component. A thermodynamic model of magma ocean crystallization (Boukaré et al., 2015) suggests that the silicate melt fraction would be gradually enriched in iron with $\text{Fe}/(\text{Fe}+\text{Mg})$ more than 0.3 in the lower mantle after 60 wt% of the melt has solidified. The $\text{Fe}/(\text{Fe}+\text{Mg})$ ratio in the residual melt remaining in the lowermost mantle could be up to 0.5 near the end of the crystallization. The amount of Fe^{3+} in this melt depends on the amount of Fe^{2+} that would disproportionate into Fe^{3+} plus Fe^0 and the efficiency of Fe^0 droplet segregation. The required bulk composition with $\text{MgSiO}_3:\text{Fe}_2\text{O}_3$ equal to 9:1 could be produced when 40-80% Fe^{2+} undergoes the disproportionation reaction and all Fe^0 migrates into the core. Fe^{3+} would be incorporated into bridgmanite with further crystallization, and our experiments and *ab initio* calculations indicate that in these Fe^{3+} -rich regions, a portion of $(\text{Mg}_{0.5}\text{Fe}_{0.5})(\text{Si}_{0.5}\text{Fe}_{0.5})\text{O}_3$ silicate would form as a separate phase, coexisting with Fe^{3+} -poor silicate. Our geodynamic modeling demonstrates that the Fe^{3+} -rich piles with ~18% $(\text{Mg}_{0.5}\text{Fe}_{0.5})(\text{Si}_{0.5}\text{Fe}_{0.5})\text{O}_3$ Bdg, which is ~1.5% intrinsically

denser than the ambient mantle (Fig. 4c), could form large-scale thermochemical structures in the lowermost mantle without being mixed into the background mantle throughout Earth's history. Please refer to the discussion in lines 228-251.

Comment 2

The main problem with the author's scenario is why would such an enormously ferric iron enriched region of the mantle ever form? If we assume that the authors are correct and there is a ferric iron enriched form of bridgmanite that coexists on the other side of a miscibility gap from MgSiO₃ bridgmanite, how did the mantle ever get so enriched in ferric iron such that the bulk composition can enter this gap? The main reason people have argued that high ferric iron contents can be obtained in the lower mantle is through charge disproportionation of Fe²⁺ to create Fe³⁺ and iron metal. The problem with this scenario is that while it raises the ferric/ferrous ratio, in the end the total iron content of silicates decreases because iron metal is formed. This process can turn a significant proportion of ferrous iron in to ferric iron but it cannot raise the ferric iron component of the mantle independently. Also disproportionation, if it occurs, is driven by the requirement of a particular phase for a raised level of ferric iron, while in equilibrium with iron metal. Once this is satisfied for normal MgSiO₃ bridgmanite, why should more ferric iron be produced to stabilise an additional ferric iron enriched bridgmanite? What process would enrich the mantle in ferric iron to a level which is in excess of what the existing mantle minerals can accommodate?

Reply:

The first question is, how much iron enrichment is so much as to be unreasonable? Given that many studies in the literature have proposed Fe-enrichment to explain the properties and possible long-term dynamic stability of the LLSVPs, we are already comfortable with the possibility of large-scale Fe-enrichment at the base of the mantle through magma ocean crystallization (Garnero et al., 2016; Labrosse et al., 2007; Lee et al., 2010; Nomura et al., 2011). For instance, according to the self-consistent thermodynamic model of magma ocean crystallization in Boukaré et al. (2015), the silicate melt fraction would be gradually enriched in iron with Fe/(Fe+Mg) > 0.3 in the lower mantle after 60 wt% of the melt solidifying. The Fe/(Fe+Mg) ratio in the residual melt remaining in the lowermost mantle could be up to 50% near the end of the crystallization. This means that our major remaining question must be, can we constrain the oxygen fugacity of the basal magma ocean and the redox state of Fe in Fe-enriched LLSVPs consistent with observed physical properties? Our results provide an important benchmark to resolve this question.

Our constraint on the **miscibility gap** is one tool we can use to determine the amount of iron enrichment required to generate disproportionation of Fe³⁺-rich bridgmanite. The composition of the Fe-poor bridgmanite in our experiments constrains the edge of the miscibility gap at pressures 15-24 GPa to ~Mg#90-92. Compared to a generally-acceptable bdg Mg#~95 in "average mantle", this suggests that relatively small enrichment in iron is sufficient to generate disproportionation, and this dense material could accumulate at the base of the mantle over Earth's history. The true Fe³⁺

amount in silicate melts associated with the basal magma ocean depends on the amount of Fe^{2+} that would disproportionate into Fe^{3+} plus Fe^0 and the efficiency of Fe^0 droplet segregation. The required bulk composition with $\text{MgSiO}_3:\text{Fe}_2\text{O}_3$ equal to 9:1 could be produced when 40-80% Fe^{2+} undergoes the disproportionation reaction and all Fe^0 migrates into the core. Fe^{3+} would be incorporated into bridgmanite with further crystallization, and our experiments indicate that in these Fe^{3+} -rich regions, a portion of $(\text{Mg}_{0.5}\text{Fe}_{0.5})(\text{Si}_{0.5}\text{Fe}_{0.5})\text{O}_3$ silicate would form as a separate phase, coexisting with Fe^{3+} -poor silicate. Due to the large excess density, $(\text{Mg}_{0.5}\text{Fe}_{0.5})(\text{Si}_{0.5}\text{Fe}_{0.5})\text{O}_3$ silicate could descend to the base of the lower mantle through mantle convection and result in Fe^{3+} -rich bridgmanite piles. Our geodynamic modeling demonstrates that the Fe^{3+} -rich piles with ~18% $(\text{Mg}_{0.5}\text{Fe}_{0.5})(\text{Si}_{0.5}\text{Fe}_{0.5})\text{O}_3$ Bdg, which is ~1.5% intrinsically denser than the ambient mantle (Fig. 4c), could form large-scale thermochemical structures in the lowermost mantle without being mixed into the background mantle throughout Earth's history. Please refer to the discussion in lines 228-251.

Comment 3

The authors also mention subduction of banded iron formations which are certainly ferric iron enriched but there are also problems with this scenario. Putting aside for the moment arguments as to whether they would indeed have formed in deep sea environments or whether such accumulations would be subducted or accreted to continents, comes the issue that the shear mass of material required i.e. 3-9 % of the entire mantle, would be far outside of estimates for how much of these formations we would expect to find. It must be born in mind that banded iron formations are produced as insoluble ferric iron is formed through reaction with oxygen produced by photo synthesis. Therefore to balance the ferric iron produced there needs to be a complementary reservoir of reduced carbon that would need to be very very significant.

Reply: We agree with the reviewer and deleted the discussion about the subduction of banded iron formations.

Comment 4

The authors also do not consider that MgSiO_3 bridgmanite can already accommodate a significant proportion of ferric iron as demonstrated in a number of studies such as those of Catalli et al. (2010) and Hummer and Fei (2012). The results of the current study are actually not discussed in light of these previous studies.

Reply:

Hummer and Fei (2012) used the multi-anvil apparatus to synthesize Fe^{3+} -only bridgmanite samples. In Table 1 of Hummer and Fei (2012), no bridgmanite sample in their study has Fe more than 3.7 mol% in each cation site, although 5.0 mol% is expected in their 'Pv10' samples. Hummer and Fei (2012) used Pt as the sample capsules, which is known to absorb Fe from Fe-bearing materials during high P - T experiments. Therefore, the bulk Fe contents in their experiments may be lower than their starting mixtures. Moreover, the coexistence of bridgmanite, MgO, and SiO_2 in

their experiments indicate their synthesis experiments were incomplete, which may cause the difference compared with our experimental observations.

Catalli et al. (2010) used laser-heated diamond anvil cells for the bridgmanite sample synthesis, which is very challenging to maintain specific P - T conditions. Although the starting material was 90 mol% MgSiO_3 –10 mol% Fe_2O_3 , there was significant loss of Fe and Mg during melting. More importantly, the homogeneity of the starting materials in tens of micrometers scale is also challenging to be reached; the redox reactions between the samples and diamond anvils during high-temperature experiments are difficult to be prevented. In addition, the lack of the chemical characterization of the recovered samples from their DAC experiments makes it difficult to verify the compositions and valence states of Fe in their bridgmanite samples.

We discussed these previous studies in lines 110-122:

"A previous multi-anvil study³⁹ synthesized Fe^{3+} -only bridgmanite with 2-4 mol% Fe^{3+} in the cation sites but did not observe the Fe^{3+} -rich Bdg phase, possibly because the bulk Fe content of their experiments is not high enough to enable the formation of such Fe^{3+} -rich Bdg. Moreover, the presence of unreacted MgO and SiO_2 in their run products (Fig. 1 in ref. ³⁹) suggests that their starting materials may not be homogenous or their experiments did not reach chemical equilibrium. Another study⁴⁰ synthesized Fe^{3+} -only Bdg with the starting material of 90 mol% MgSiO_3 –10 mol% Fe_2O_3 using laser-heated diamond anvil cell (LH-DAC)⁴⁰. However, the chemical composition of Fe^{3+} -only Bdg was not reported, possibly because there was a significant loss of Fe and Mg during melting⁴⁰ and some Fe^{3+} was reduced through reaction with diamond during laser heating⁴¹. In addition, the proportion of the Fe-rich Bdg phase is much smaller than the Fe-poor Bdg (Fig. 1), and therefore it is difficult to detect without a detailed analysis of the run products in ref. ⁴⁰."

Comment 5

The most important results from this study are those of the elastic properties of the ferric iron enriched phase. However apart from a couple of diagrams we get very few details on these properties. What are the individual elastic constants for example? What are the room pressure and temperature moduli? There is no evaluation of these results either. Surely with an ab initio calculation of the structure it should be possible to assess why the shear modulus is affected so strongly by the coupled substitution of ferric iron. Also the effect on the bulk modulus of ferric iron coupled substitution could be compared with the results of Catalli et al. (2010).

Reply:

Owing to the enormous computational cost of calculating the vibrational density of state based on LDA+U, we used the 20-atom unit cell for $(\text{Mg}_{0.5}\text{Fe}_{0.5})(\text{Si}_{0.5}\text{Fe}_{0.5})\text{O}_3$ bridgmanite. In this work, we only report results for aggregate elastic moduli, not individual elastic coefficients. The latter are sensitive to atomic configurations and therefore to supercell size, which can accommodate different configurations for the same composition. The aggregate elastic moduli, K and G, are quite insensitive to the

atomic configuration, which has been verified in previous studies (Nunez-Valdez et al., 2010, 2012a, 2012b, 2013; Marcondes et al., 2015). This strategy was also adopted in previous theoretical studies on the elastic properties of Fe-bearing bridgmanite (Shukla et al., 2015, 2016) with the same method as the one used in this study. Please refer to the discussion in lines 529-536.

Because $(\text{Mg}_{0.5}\text{Fe}_{0.5})(\text{Si}_{0.5}\text{Fe}_{0.5})\text{O}_3$ bridgmanite is only stable at > 23 GPa (Liu et al., 2018), the elastic moduli and sound velocities at > 20 GPa of $(\text{Mg}_{0.5}\text{Fe}_{0.5})(\text{Si}_{0.5}\text{Fe}_{0.5})\text{O}_3$ bridgmanite are shown in Fig. S2. There is no elastic data available for $(\text{Mg}_{0.5}\text{Fe}_{0.5})(\text{Si}_{0.5}\text{Fe}_{0.5})\text{O}_3$ bridgmanite in the literature to compare with our calculated results.

Although *ab initio* calculations only require the structure of bridgmanite, it is hard to explain why the shear modulus is affected so strongly by the incorporation of 50% Fe_2O_3 into bridgmanite. This trend was also found in previous theoretical studies. Shukla et al. (2016) used the same method to investigate the elastic properties of Fe^{3+} -bearing bridgmanite and found that inclusion of 12.5 mol% of Fe_2O_3 in bridgmanite changes K_S and G by $\sim 5\%$ and -15% , respectively. These results have been compared with previous experimental data including the results of Catalli et al. (2010). This study focuses on $(\text{Mg}_{0.5}\text{Fe}_{0.5})(\text{Si}_{0.5}\text{Fe}_{0.5})\text{O}_3$ bridgmanite. To the best of our knowledge, except for volumes at ambient temperature, there is no experimental data available in the literature to compare with our calculated results.

Comment 6

There are only very vague comparisons made with the expected properties of LLSVPs. The expected range of LLSVP properties should be plotted in figures 3 and 4. What is a suitable criterion for a good agreement? Some assessment of this would be required. LLSVPs do not show up significantly in terms of the bulk sound velocity which provides a further criterion.

Reply:

In recent decades, the existence of the LLSVPs was indicated by many mantle shear wave velocity tomography models, as summarized in Garnero et al. (2016). Different shear-wave tomography models agree that the shear-wave velocity anomaly ($\text{dln}V_S$) ranges from -0.5% to -1% at the shallow part of LLSVPs, while $\text{dln}V_S$ found in the bottom part could be up to -3% . The compressional-wave tomography models also reveal negative anomalies of compressional-wave velocities (V_P) (Wang et al., 2007; Frost and Rost, 2014), however, wide variations in the amplitude of V_P anomaly, the appearance shape, and the geographical location exist between different models (Garnero et al., 2016). Overall, the V_P anomaly has a smaller amplitude relative to the V_S anomaly, causing an unexpected large $\text{dln}V_S/\text{dln}V_P$ ratio (> 2) (Wang et al., 2007). We have plotted the range with a $\text{dln}V_S/\text{dln}V_P$ ratio > 2 in Fig. 4 but cannot emphasize the $\text{dln}V_S$ and $\text{dln}V_P$ because they have wide ranges. Accordingly, we focus on whether the model with enrichment of $(\text{Mg}_{0.5}\text{Fe}_{0.5})(\text{Si}_{0.5}\text{Fe}_{0.5})\text{O}_3$ bridgmanite can reproduce the ranges of $\text{dln}V_S$ and $\text{dln}V_P$ observed in the LLSVPs and the $\text{dln}V_S/\text{dln}V_P$

ratio of > 2 . The bulk sound velocity is estimated based on the V_P and V_S , which is not an independent criterion. We clarified the seismic observations in lines 24-33.

Comment 7

Lines 165 and 168 there is no explanation for the difference between these two assessments- exactly the same comparison is made with two different answers.

The y axis in figure 4 is very difficult to understand. Why is the difference between Fe_{2+Fp} and $Fe_{2+Fp,pyrolite}$?

Reply:

We revised the typo in line 168. The chemical composition is $(Mg_{0.5}Fe_{0.5})(Si_{0.5}Al_{0.5})O_3$.

Previous studies (Wu, 2016) have suggested that the pyrolitic lower mantle that consists of 15 % $Mg_{0.82}Fe_{0.18}O$ ferropericlasite (Fp), 78% $Mg_{0.92}Fe_{0.08}SiO_3$ bridgmanite (Fe^{2+} -Bdg), and 7% $CaSiO_3$ Ca-perovskite (Ca-Pv) can admirably reproduce the velocity and density profiles of PREM model for the lower mantle. Based on the pyrolite model, we quantify the dependences of velocity and density anomalies on the amount of Fe^{3+} -rich Bdg by substituting a certain proportion of Fe^{3+} -free Bdg with $(Mg_{0.5}Fe_{0.5})(Si_{0.5}Al_{0.5})O_3$ bridgmanite. We also explore the effect of Fe^{2+} content in the assemblage (noted by the Fe^{2+} content in Fp, Fe^{2+}_{Fp}) on modeling results because the incorporation of Fe^{2+} into Fp and Bdg decreases their velocities to some extent. The Fe-Mg partition coefficient between Fp and Bdg is used to constrain their Fe^{2+} contents. $Fe^{2+}_{Fp, pyrolite}$ refers to the FeO content of Fp (18 mol%) in the pyrolite model for the normal lower mantle. This value is taken as a reference to show the variation of Fe^{2+} in the modeling chemical heterogeneities compared to the normal lower mantle. We change $Fe^{2+}_{Fp, pyrolite}$ to $Fe^{2+}_{Fp, NM}$ to avoid ambiguity.

Reviewer #3

Comment 1

The paper by Wang et al. attempts to unravel the origin of LLSPVs by combining high-pressure experiments with ab-initio calculations and geodynamic modelling. Understanding the origin of LLSPVs is a pivotal question in geophysics and the present study is thus very timely; it will certainly be of interest to the community.

The present manuscript proposes that iron-rich bridgmanite forms in the lower mantle and causes the notable seismic features of large low shear velocity provinces. Iron enrichment generally leads to lower seismic wave velocities and an increase in density for mantle and it is thus does not seem surprising that a model containing iron-rich bridgmanite (when mixed with variable amounts of “normal” mantle) can explain the seismic features of LLSPVs. The novelty of this paper is to show that iron-rich bridgmanite might indeed exist in the lower mantle. To reach this conclusion the paper relies on experimental work as well as ab-initio calculations.

Reply: Thanks for recognizing our contribution.

Comment 2

The authors report results from a single multi-anvil experiment at lower mantle conditions (at 24GPa, 1873K). In this experiment, they observed co-existing Fe-poor bridgmanite and Fe-rich akimotoite. They conclude that the observed akimotoite back-transformed from bridgmanite. But is there really a stable assemblage of Fe-rich and Fe-poor bridgmanite during the experiment (and in the lower mantle) or is the observed exsolution a result of the experimental procedure, i.e. a result of experimental conditions too close to the akimotoite stability field and/or a quenching result? I feel that more experiments at lower mantle conditions would be helpful to reach a robust conclusion. Also, how does the results depend on pressure. Will the exsolution process happen in the deeper mantle as well? In-situ diamond-anvil cell experiments could be helpful since they would allow for a direct observation of stable phase assemblages at deep lower mantle conditions. I note that the authors have performed *ab-initio* calculations to confirm their experimental findings and show that the Gibbs Free Energy of an assemblage of Fe-rich/Fe-poor bridgmanite is energetically favorable as opposed to a solid solution. However, the experimental verification seems weak to me.

Reply:

We ran two multi-anvil experiments in the bridgmanite (Bdg) stability field (Fig. 1 and Supplementary Table 1). In both experiments, the iron-poor phase adopts the perovskite structure, which was recovered at 1 bar and confirmed by Raman spectroscopy measurements. Our in-situ XRD results show that the Fe-rich phase undergoes reversible transformation between the akimotoite and perovskite structures at 24 GPa and 300 K (Supplementary Figure 1), and therefore the Fe-rich phase at the *P-T* conditions of the two experiments should also adopt the perovskite structure, even though it back-transformed into the akimotoite structure upon decompression. The grain size and distribution of the products from the two 24-GPa experiments indicate that the coexistence of Fe-rich and Fe-poor bridgmanite phases is not a result of quenching exsolution. The samples were quenched from 1873 K to 573 K in 2 seconds. It is unlikely that such fast quenching of a solid phase would produce unevenly distributed Fe-rich grains that are as large as 20-micron in size.

Our previous DAC experiments confirmed that the Fe³⁺-rich Bdg remains stable to much greater depths in the lower mantle (Liu et al. 2018, *Nature Communications*). In-situ diamond-anvil cell experiments can extend the *P-T* regions for studying the chemical reactions in the MgSiO₃-Fe₂O₃ system. However, it is challenging to control the oxygen fugacity and determine the chemical compositions and Fe valence states of the recovered phases. We hope that our findings will motivate further investigations using DAC or multi-anvil apparatus with sintered diamonds.

As an alternative solution, we performed *ab initio* calculations to investigate the stability of (Mg_{0.5}Fe_{0.5})(Si_{0.5}Fe_{0.5})O₃ Bdg under lower-mantle conditions. The results show that the assemblage of (Mg_{0.5}Fe_{0.5})(Si_{0.5}Fe_{0.5})O₃ and MgSiO₃ Bdg has a lower Gibbs free energy than a single-phase (Mg_{0.875}Fe_{0.125})(Si_{0.875}Fe_{0.125})O₃ Bdg under the *P-T* of the whole lower mantle regardless of the spin state (Supplementary Fig. 3),

indicating that the mixed two phases are more stable than the single-phase Bdg with a homogeneous composition. Our theoretical results support our experimental observations and reveal that under lower-mantle conditions, this Fe³⁺-rich Bdg with the chemical composition of approximately (Mg_{0.5}Fe_{0.5})(Si_{0.5}Fe_{0.5})O₃ should form as a separate phase coexisting with Fe-poor Bdg in the bulk composition of 90 mol% MgSiO₃–10 mol% Fe₂O₃ due to the miscibility gap.

Comment 3

Both experiments and calculations are reported for a pure MgSiO₃-Fe₂O₃ system. How does the system change when considering a bulk lower mantle composition, i.e. does the presence of ferropervicite and CaPv perovskite change the stable phase assemblage? Also, how does Al-incorporation (that is expected in the lower mantle) change the equilibrium phase assemblage?

Reply:

As CaPv perovskite adopts a different structure compared with bridgmanite and there is a big miscibility gap between MgSiO₃ Bdg and CaSiO₃ perovskite CaSiO₃ (e.g., Armstrong et al., 2012; Irifune et al., 1989; Jung and Schmidt, 2011; Tamai and Yagi, 1989), the presence of CaPv perovskite should have a minor effect on the phase diagram of the MgSiO₃-Fe₂O₃ system. Although a recent laser-heated diamond anvil cell study (Grease et al., 2020) proposes that Ca could enter Mg-dominant bridgmanite, however as states by Grease et al. (2020), the mechanism and extent for incorporating Ca in Mg-dominant bridgmanite are still unclear. Similarly, very limited Fe³⁺ can be incorporated into Fp, and therefore, Fp should not affect the phase relationship reported in this study. Some Al can be incorporated into bridgmanite and it will compete for the B-site of bridgmanite with Fe³⁺. Our recent work successfully synthesized (Mg_{0.5}Fe³⁺_{0.5})(Al_{0.5}Si_{0.5})O₃ samples (Zhu et al., 2020, JGR, 125) in the bulk composition with MgSiO₃:Fe₂O₃:Al₂O₃=2:1:1. This implies that the presence of Al would not prevent the formation of very Fe³⁺-rich bridgmanite. This study focuses on Fe³⁺-rich materials with a much higher Fe³⁺/Al ratio. We can expect such a chemical heterogeneity will produce (Mg_{0.5}Fe_{0.5})(Si_{0.5}Fe_{0.5})O₃ bridgmanite as observed in this study, in which a small amount of Al may enter into the B site of bridgmanite if there is some Al₂O₃ in this system. We also consider the Al effect on the modeling results of LLSVPs' observations, as shown in Fig. 3 and Fig. 4.

Comment 4

According to the reported findings, about 15% of (Mg_{0.5}Fe_{0.5})(Si_{0.5}Fe_{0.5})O₃ bridgmanite is required to explain the seismic features of the LLSVPs. This sounds quite a lot to me. Can they authors provide some rough calculations to show that the required Fe³⁺ could indeed originate from the disproportionation of Fe²⁺ in an early magma ocean as suggested in the manuscript (motivated by recent high-pressure experiments, again at shallow lower mantle conditions (Armstrong et al. 2019))?

Reply:

The present model for chemical heterogeneities within the LLSVPs is consistent with

Fe-rich remnants of a basal magma ocean created early in Earth's history^{7,24–26}. Ferrous iron in dense silicate melts associated with the basal magma ocean would partially disproportionate to Fe^{3+} plus Fe^0 at high pressures³⁰ and segregation of precipitated Fe^0 from the magma ocean into core would enrich silicate melt in Fe_2O_3 component. A thermodynamic model of magma ocean crystallization⁵³ suggests that the silicate melt fraction would be gradually enriched in iron with $\text{Fe}/(\text{Fe}+\text{Mg})$ more than 0.3 in the lower mantle after 60 wt% of the melt has solidified. The $\text{Fe}/(\text{Fe}+\text{Mg})$ ratio in the residual melt remaining in the lowermost mantle could be up to 0.5 near the end of the crystallization. The amount of Fe^{3+} in this melt depends on the amount of Fe^{2+} that would disproportionate into Fe^{3+} plus Fe^0 and the efficiency of Fe^0 droplet segregation. The required bulk composition with $\text{MgSiO}_3:\text{Fe}_2\text{O}_3$ equal to 9:1 could be produced when 40-80% Fe^{2+} undergoes the disproportionation reaction and all Fe^0 migrates into the core. Fe^{3+} would be incorporated into bridgmanite with further crystallization, and our experiments and *ab initio* calculations indicate that in these Fe^{3+} -rich regions, a portion of $(\text{Mg}_{0.5}\text{Fe}_{0.5})(\text{Si}_{0.5}\text{Fe}_{0.5})\text{O}_3$ silicate would form as a separate phase, coexisting with Fe^{3+} -poor silicate. Due to the large excess density, $(\text{Mg}_{0.5}\text{Fe}_{0.5})(\text{Si}_{0.5}\text{Fe}_{0.5})\text{O}_3$ silicate could descend to the base of the lower mantle through mantle convection and result in Fe^{3+} -rich bridgmanite piles. Our geodynamic modeling demonstrates that such Fe^{3+} -rich piles with ~18% $(\text{Mg}_{0.5}\text{Fe}_{0.5})(\text{Si}_{0.5}\text{Fe}_{0.5})\text{O}_3$ Bdg, which is ~1.5% intrinsically denser than the ambient mantle (Fig. 4c), could form large-scale thermochemical structures in the lowermost mantle without being mixed into the background mantle throughout Earth's history
Please refer to the discussion in lines 228-251.

Comment 5

The authors have done calculations of the elastic constants (C_{ij}) of Fe-bridgmanite and Fe/Al-bridgmanite. The results are of general interest to the geophysics community and should be tabulated along with the bulk elastic properties. It would also be nice to see more careful comparison of the results with previous calculations and available experimental measurements of wave velocities (acknowledging limitations in the compositional and pressure/temperature ranges that have been studied), some that come to mind include Kurnosov et al. 2017, Fu et al. 2018, Fu et al. 2019.

Reply:

Owing to the enormous computational cost of calculating the vibrational density of state based on LDA+U, we used the 20-atom unit cell for $(\text{Mg}_{0.5}\text{Fe}_{0.5})(\text{Si}_{0.5}\text{Fe}_{0.5})\text{O}_3$ bridgmanite. In this work, we only report results for aggregate elastic moduli, not individual elastic coefficients. The latter are sensitive to atomic configurations and therefore to supercell size, which can accommodate different configurations for the same composition. The aggregate elastic moduli, K and G, are quite insensitive to the atomic configuration, which has been verified in previous studies (Nunez-Valdez et al., 2010, 2012a, 2012b, 2013; Marcondes et al., 2015). This strategy was also adopted in previous theoretical studies on the elastic properties of Fe-bearing bridgmanite (Shukla et al., 2015, 2016) with the same method as the one used in this study. Please

refer to the discussion in lines 529-536.

This study focuses on the elastic properties of $(\text{Mg}_{0.5}\text{Fe}_{0.5})(\text{Si}_{0.5}\text{Fe}_{0.5})\text{O}_3$ bridgmanite. To the best of our knowledge, except for volumes at ambient temperature, there is no experimental data available in the literature to compare with our calculated results. None of the previous studies mentioned by the reviewer including Kurnosov et al. (2017), Fu et al. (2018), and Fu et al. (2019) have investigated the properties of $(\text{Mg}_{0.5}\text{Fe}_{0.5})(\text{Si}_{0.5}\text{Fe}_{0.5})\text{O}_3$ bridgmanite. Shukla et al. (2016) used the same method as this study to investigate the elastic properties of $(\text{Mg}_{0.875}\text{Fe}_{0.125})(\text{Si}_{0.875}\text{Fe}_{0.125})\text{O}_3$ bridgmanite and found that the inclusion of 12.5 mol% of Fe_2O_3 in bridgmanite changes K_S and G by $\sim 5\%$ and -15% , respectively. These results have been compared with previous experimental data including the results of Catalli et al. (2010) in Shukla et al. (2016). In addition, the theoretical results of Shukla et al. (2016) have also been compared and discussed by Kurnosov et al. (2017) and Fu et al. (2018).

Comment 6

There are some typos throughout the manuscript, e.g. “origin” in line 45, “reported” in line 91, “corresponding” in line 107, “bulk modulus” in line 157,.. Please check and correct.

Reply: We carefully checked and corrected all typos in the revised manuscript.

Comment 7

In summary, the paper presents interesting findings with possibly important consequences. Unfortunately, the conclusions fundamentally rely on the assumption that iron-rich bridgmanite forms as stable phase in the lower mantle, and that it exists a sufficiently large amount. At this point, I am not convinced that the experimental results presented in the paper provide robust evidence for this process to happen in the lower mantle. Having said that, it is worth re-iterating the results are of significant interest and I am convinced that they will trigger follow-up research to test the here-hypothesized nature of LLSVPs.

Reply:

Thanks for praising our contribution. Our experimental results are reliable (see the replies above) and our theoretical calculations also strongly support our observations, and will stimulate further studies extending investigation of the $\text{MgSiO}_3\text{-Fe}_2\text{O}_3$ phase diagram to greater depths in the mantle. The present model for chemical heterogeneities within the LLSVPs can be derived from the disproportionation of Fe^{2+} into Fe^{3+} plus Fe^0 in the early magma ocean. The magma ocean crystallization model suggests that the silicate melt fraction would be gradually enriched in iron with $\text{Fe}/(\text{Fe}+\text{Mg}) > 30\%$ in the lower mantle after 60 wt% of the melt solidifying. The required Fe^{3+} -rich heterogeneity could be produced when 40-80% Fe^{2+} undergoes the disproportionation reaction and all Fe^0 migrates into the core. Fe^{3+} would be incorporated into bridgmanite with further crystallization, and our experiments indicate that in these Fe^{3+} -rich regions, a portion of $(\text{Mg}_{0.5}\text{Fe}_{0.5})(\text{Si}_{0.5}\text{Fe}_{0.5})\text{O}_3$ silicate would form as a separate phase, coexisting with Fe^{3+} -poor silicate. Our geodynamic

modeling demonstrates that the Fe^{3+} -rich piles with $\sim 18\%$ $(\text{Mg}_{0.5}\text{Fe}_{0.5})(\text{Si}_{0.5}\text{Fe}_{0.5})\text{O}_3$ Bdg could form large-scale thermochemical structures in the lowermost mantle without being mixed into the background mantle throughout Earth's history. Please see our replies above.

REVIEWERS' COMMENTS

Reviewer #1 (Remarks to the Author):

The revised version of the paper looks better. The authors have now shown that the predicted stability of $(\text{Mg}_{0.5}\text{Fe}_{0.5})(\text{Si}_{0.5}\text{Fe}_{0.5})\text{O}_3$ bridgmanite is insensitive to the choice of exchange-correlation functional (LDA/GGA).

Here are a few points for them to consider and comment:

- a) It would be still helpful to show the calculated pressure-volume values along 300 K curve or may be static (0 K) results in SI.
- b) The authors should mention about a possible explanation in terms of FeO_2Hx system for LLSVPs as proposed by Deng et al., JGR, 2019.
- c) Fig 2: Since you have now done some GGA+U calculations, could you show those P-V results in the plot. It would be interesting to see how LDA and GGA P-V results compare with the experimental data.

Towards comments raised by ref #2 and authors' revisions:

The bottommost part of the lower mantle (e.g., LLSVPs) is extremely complex region both geo-chemically and physically. The paper by Wang et al. propose one plausible explanation for the origin of LLSVPs based on their measured and computational results.

The authors' response to the comments/suggestions of Reviewer # 2 looks mostly fine. A regional enrichment in iron to produce ferric iron-rich bridgmanite at the bottom of lower mantle is possible because of a dense residual basal magma ocean and its solidification. The revision version of the paper also reads well.

Their analysis and discussion are based on aggregate wave velocities, which require the knowledge of bulk (K) and shear (G) moduli for iron-rich bridgmanite systems. It is still helpful to present a full set of calculated results of individual elastic constants in supplementary materials. One scenario to consider is how the predicted low shear modulus G could be associated with individual shear elastic coefficients (c_{44} , c_{55} and c_{66}) and tetragonal shear moduli (e.g., $c_{11}-c_{12}$, $c_{22}-c_{23}$, etc.).

Reviewer #3 (Remarks to the Author):

Review of REVISED Nature Communications Manuscript "Formation of large low shear velocity provinces through decomposition of oxidized mantle" by Wenzhong Wang et al.

Dear editor, dear authors,

I only have time for a very brief review on the revised manuscript submitted by Wang et al., my apologies for that. I hope that it can be of help.

In my original review, I raised three major concerns regarding (1) some weaknesses in the experimental results, (2) possible effects of chemical complexity, and (3) the availability of a sufficient amount of $(\text{Mg}_{0.5}\text{Fe}_{0.5})(\text{Si}_{0.5}\text{Fe}_{0.5})\text{O}_3$ bridgmanite, i.e. total iron, to quantitatively explain seismic features of the LLVSPs. First of all, I would like to thank the authors for addressing all of these concerns in their reply.

After reading the authors' reply, I am broadly convinced by the authors response to concern #2. My concern #1 has been somewhat addressed by the author's reply and the additions made to the manuscript, but I still have some doubts about the conclusions drawn from two experiments and feel that further experiments would help. However, the ab-initio calculations add important support to the drawn conclusions, but I am not qualified to judge the quality of these (but I realized that one of the other reviewers commented in more detail on these)

I still left with some doubts about the mass balance calculations and the possible availability of sufficient (18%) of $(\text{Mg}_{0.5}\text{Fe}_{0.5})(\text{Si}_{0.5}\text{Fe}_{0.5})\text{O}_3$ bridgmanite in the lowermost mantle as a result of magma ocean crystallization. Would some calculations/graphs showing the approximate of amount Fe^{3+} -rich bridgmanite (and Fe^{2+} rich ferropericlase) with progressing crystallization of the magma ocean or similar be helpful to address this concern and convincingly show that the proposed scenario is realistic?

Reviewer #4 (Remarks to the Author):

In this manuscript, the authors provide a novel idea for explaining LLVSPs via enrichment of high ferric Fe bridgmanite in these areas. The high ferric Fe forms due to a miscibility gap when you have intermediate Fe-bearing bridgmanite. The authors provide new, solid experimental and computational support for this scenario. I also read through the lengthy and thorough response to the reviewers. All three previous reviewers as well as I agree that the idea is interesting and appropriate for the broad readership of Nature Communications. Where there are concerns (especially from Reviewer 2) is regarding whether the authors have provided strong enough evidence to support their idea, especially in light of other hypotheses for explaining LLVSPs. Reviewer 2's criticism can be leveled on many highly cited and high profile papers studying anomalies at the base of the lower mantle. This is a challenge of deep Earth studies – we do not have direct samples of the deepest parts of Earth's mantle or other strict constraints on the composition in these regions, so current seismic constraints can be matched by different combinations of varying composition, proportions of different phases, temperature, etc. As a broader community we need to continue to improve and expand our observational constraints, and also continue to develop experimental and computational tools. What I really like about this paper and why I agree with Reviewers 1 & 3, is that it is state of the art for what we can do now, and the new hypothesis introduced by the authors will certainly generate significant interest and discussion in the community, and stimulate future progress in the field.

Manuscript NCOMMS-20-38791a – Response to the Reviewers' comments

We are grateful to the Reviewers for providing valuable feedback and comments to our original manuscript and thank them for recommending the revised manuscript for publication in Nature Communications.

Reviewer #1 (Remarks to the Author):

I have carefully looked over the revised draft, as well as the authors' point-by-point response. As I said in my first review, I believe this manuscript will be of high interest to the community, and it represents a new step forward that is likely to inspire advances in the design of nanoscale active matter systems. The authors' have addressed all of my concerns with the manuscript, and I recommend it for publication with no reservations.

We thank the reviewer for his/her original comments and carefully reading our revised version of the manuscript.

Reviewer #2 (Remarks to the Author):

The authors have addressed the reviewers' comments satisfactorily. Therefore, I recommend its publication in Nature Communication.

We are grateful for his/her valuable feedback that helped us to improve our manuscript and thank him/her for recommending it for publication in Nature Communication.

Reviewer #3 (Remarks to the Author):

I believe there are some interesting findings from this paper that could be useful to the optical nano-manipulation community. I appreciate the additional section to introduce more details to this paper. I am still of the opinion that the original contributions of this paper are unsatisfactory but if the editors and other reviewers disagree then I don't feel the need to make a strong stand on this point.

We thank the reviewer for his/her interest in the paper and in the additional information we have made to provide a better picture of the novelty of our work.